



# OMI Collection 4: establishing a 17-year long series of detrended L1b data

Quintus Kleipool[1], Nico Rozemeijer[1,2], Mirna van Hoek[1], Jonatan Leloux[1,2], Erwin Loots[1], Antje Ludewig[1], Emiel van der Plas[1], Daley Adrichem[1,2], Raoul Harel[1,2], Simon Spronk[1,2], Mark ter Linden[1,3], Glen Jaross[4], David Haffner[4], Pepijn Veefkind[1], and Pieternel Levelt[5]

[1]Royal Netherlands Meteorological Institute (KNMI), De Bilt, The Netherlands
[2]TriOpSys B.V., Utrecht, The Netherlands
[3]Science and Technology (S[&]T), Delft, The Netherlands
[4]NASA Goddard Space Flight Center (GSFC), Greenbelt, Maryland, United States
[5]National Center for Atmospheric Research (NCAR), Boulder, Colorado, United States

**Correspondence:** Antje Ludewig (antje.ludewig@knmi.nl)

**Abstract.** The Ozone Monitoring Instrument (OMI) was launched on July 15, 2004, with an expected mission lifetime of 5 years. After more than 17 years in orbit the instrument is still functioning satisfactorily, and in principle can continue doing so for many years more. In order to continue the datasets acquired by OMI and the Microwave Limb Sounder the mission was extended up to at least 2023.

Actions have been taken to ensure the proper functioning of the OMI instrument operations, the data processing, and the calibration monitoring system until the eventual end of the mission. For the data processing a new level 0 to level 1b data processor was built based on the recent developments for Tropospheric Monitoring Instrument (TROPOMI). With corrections for the degradation of the instrument now included, it is feasible to generate a new data collection to supersede the current collection 3 data products.

This paper describes the differences between the collection 3 and collection 4 data. It will be shown that the collection 4 L1b data is a clear improvement with respect to the previous collections. By correcting for the gentle optical and electronic aging that has occurred over the past 17 years, OMI's ability to make trend-quality ozone measurements has further improved.

## 1 Introduction

The OMI is a space-borne nadir viewing imaging spectrometer with two separate channels that measure the solar radiation
scattered back by the Earth's atmosphere and surface over the entire wavelength range from 270 to 500 nm with a spectral resolution of about 0.5 nm. OMI is on the National Aeronautics and Space Administration's (NASA) Earth Observing System (EOS) Aura satellite (Schoeberl et al., 2006), together with the Microwave Limb Sounder (MLS), the Tropospheric Emission Spectrometer (TES) and the High Resolution Dynamics Limb Sounder (HIRDLS).

The objective of OMI is measuring a number of trace gases in both the troposphere and stratosphere in a high spectral
and spatial resolution (Levelt et al., 2006). The heritage of OMI are the European ESA instruments GOME and SCIA-MACHY (Bovensmann et al., 1999), which introduced the concept of measuring the complete spectrum in the ultraviolet,



visible and near-infrared wavelength range with a high spectral resolution. This enables the retrieval of several trace gases from the same spectral measurement. The American predecessors of OMI are the NASA SBUV (Cebula et al., 1988) and TOMS (McPeters et al., 1998) instruments. TOMS measured 8 wavelength bands from which the ozone column was obtained,

but had the advantage that it had a fairly small ground-pixel size (50 km × 50 km) combined with daily global coverage. OMI combines the advantages of GOME and SCIAMACHY with those of TOMS, allowing the measurement of the complete spectrum in the ultraviolet/visible wavelength range with a very high spatial resolution *and* daily global coverage. This was made possible by using a novel optical design using two-dimensional charge coupled device (CCD) detectors in the focal plane area.

The OMI instrument operates in a push-broom configuration with a wide swath. The 115° viewing angle of the telescope together with a polar circular orbit of about 705 km altitude corresponds to a 2600 km wide swath on the Earth's surface, which allows OMI to achieve daily coverage of the complete Earth. Light from the entire swath is recorded simultaneously and dispersed by gratings onto one direction of the two-dimensional detectors. The spectral information for each position is projected onto the other direction of the detectors.

The obtained spectra are used to retrieve the primary data products: $O_3$ total column, $O_3$ vertical profile, UV-B flux, $NO_2$ total column, aerosol optical thickness, effective cloud cover and cloud top pressure. In addition the following secondary data products are retrieved: $SO_2$ total column, BrO total column, $CH_2O$ total column and $ClO_2$ total column.

The Sun irradiation is measured on a daily basis via a dedicated solar port. When this data is used to calculate the observed Earth's reflectance, the small variability of the solar output is compensated. In addition, instrumental effects that are common

to the Earth and Sun port are effectively eliminated as well.

## 1.1 Instrument description

The OMI instrument has already been described in detail in previous publications (Dobber et al., 2006; Schenkeveld et al., 2017). In addition to these papers, a rewritten description of the instrument and the collection 4 L01b processing algorithms can be found in the ATBD (Ludewig et al., 2021), provided as supplemental information to this paper.

Noteworthy is the fact that the instrument uses two CCD detectors referred to as the UV and VIS detectors, respectively. The VIS detector is used for a single continuous wavelength range, the UV detector, however, is used for two separate wavelength ranges, referred to as UV1 and UV2, respectively. This naming convention was used throughout the mission up to and including collection 3. For collection 4 the TROPOMI naming convention was adopted, referring to the UV1, UV2 and VIS channels as band 1, band 2 and band 3 respectively. In this paper both conventions are used interchangeably.

## 50 1.2 History of OMI L1b data collections

The integration of the OMI proto-flight model was completed in 2001 and followed by the on-ground calibration campaign which took place from April to November 2002. This calibration was performed partly in ambient and partly under flight representative conditions in thermal vacuum. Pre-flight calibration parameters were retrieved from these calibration measurements





and used for the collection 1 dataset, which was active until 25 March 2005. The on-ground calibration campaign is described
in detail in Dobber et al. (2006).

Re-analysis of on-ground measurements lead to replacements of radiometric key data. Furthermore the straylight calibration
key data were updated, the gain of the detector electronics modules and the dark current were updated, and the non-linearity
threshold was increased. The resulting calibration characteristics of this effort are also described in Dobber et al. (2006). These
updates constitute the collection 2 version that became active in forward processing on 26 March 2005.

Collection 3 covers a major update of the L01b processor software and the corresponding calibration key data. Insights
from in-flight measurements and comparison with models served as input for this update. The papers by Dobber et al. (2008a)
and Dobber et al. (2008b) describe these updates in great detail. This version was activated in reprocessing mode on March
2007, generating collection 3 data for the mission so far. The reprocessing caught up with the forward processing mode in
November 2007, from which point onwards the collection 3 was fully active, and the generation of collection 2 data was
discontinued.

## 1.3 Inflight calibration monitoring

The inflight trend and calibration monitoring system (TMCF) was developed at KNMI, prior to launch, to ensure that the
calibration status of the instrument could be monitored during the mission.

After launch, it was found that radiation damage occurred to the CCD detectors due to the impact of high energy protons
and electrons from the cosmic environment. This degradation may result in increased pixel dark current in the event of a hit.
The pixel dark current is corrected for by the L01b processor, however, no temporal aspect of this correction was anticipated.

In addition it was observed that the dark current could also become unstable at unpredictable timescales, an effect know as
random telegraph signal (RTS). Because RTS prevents accurate correction for the dark current, it was decided to identify these
situations, and to flag pixels accordingly when necessary using a continuously updated dead and bad pixel map.

For collection 3, the TMCF system was upgraded such that it could be used in a closed-loop with the OMI science
investigator-led processing system (OSIPS) in the NASA ground segment. The TMCF analyzes the calibration data produced
at the NASA OSIPS on a daily basis, calculates dynamic dark current maps and RTS maps, and sends this updated calibration
key data back to the OSIPS to be used in the forward processing.

Since June 2007 OMI has suffered from the so-called row-anomaly (RA) phenomenon, where certain cross-track field-of-
views (rows) are seemingly blocked resulting in abnormally low radiance readings. The most probable cause of blocking is a
partial external obscuration of the radiance port by a piece of loose multi-layer insulation (MLI) of the instrument itself, but
this is not certain. Up to now, the row anomaly remains elusive, and continuously evolves over time, in both the number of rows
that are affected, as well how the phenomenon manifests itself in the measurements. Data from the TMCF system is analyzed
to determine which rows are affected and should be flagged accordingly. Updates to the row anomaly flagging are also handled
by the TMCF / OSIPS sytem, but only after human inspection.



In 2016 an analysis was done of all available TMCF data for the first 12 years of the mission as described by Schenkeveld et al. (2017). Other than some software bug fixes and the updates to the row anomaly flagging scheme, no further changes were made to the collection 3 calibration.

## 1.4 Rationale for collection 4

The OMI instrument was designed with an anticipated lifetime of 5 years, but after 17 years in orbit the instrument is still functioning properly. Even though OMI can continue working for many years, the Aura satellite must meet the 25-year re-entry requirement, and therefore needs to have enough fuel left to exit the A-train constellation. This means that there are only a few inclination adjust maneuvers (IAMs) that can still be performed. After the last IAM is performed, the satellite local time ascending node (LTAN) crossing will drift towards a time later in the afternoon. Aura could leave the constellation as early as

mid 2024 when the LTAN has drifted up to 14h00 or as late as July 2025. After Aura has exited the A-train, the power budget from the solar panels becomes the limiting factor, and it is estimated that the mission must end in August 2025.

In order to ensure that the OMI can remain functioning for this extended time period under all aforementioned changes, actions were taken with respect to instrument operations, data processing and the calibration monitoring system. Due to the fact that the TMCF system reached end-of-live in late 2020 a major overhaul of the data processing system was needed. The

best solution was to create a new L0 to L1b data processor for OMI, based on the available TROPOMI L01b development. This updated OMI processor has in-orbit calibration functionality in forward mode, making the TMCF system obsolete. The available TMCF calibration data has been analyzed, such that historic trends in the instrument calibration status can be corrected for in the collection 4 L01b (re-)processing. This will result in the collection 4 OMI L1b data product that resembles the TROPOMI (Veefkind et al., 2012) data product as much as possible using modern data formats and metadata definitions.

In addition, both the processor and instrument operations are optimized such that the data processing system is robust against changes in the instrument, its operations, and the orbital parameters of the satellite, which allows stable operations until the end of the Aura mission.

With the efforts described in this paper a 17-year data record of Earth spectral reflectances is established that has been corrected as far as possible for trends and degradation that occurred during the lifetime of the instrument. The processing

system is designed to facilitate this de-trending until the end of the mission, at which point OMI will have a data record of over two decades. Due to the upgrades to the file format, this OMI series can be readily connected and combined with the data series of its successor TROPOMI.

In parallel to the L1b development described in this paper, all KNMI and NASA L2 processors are also updated. The KNMI L2 updates are based on the most recent L2 processors in use for TROPOMI, but these developments fall outside the scope of

this paper.

## 1.5 Outline

After this introduction, the paper continues in Sect. 2 with an overview of the changes made to the instrument operations baseline needed to guarantee stable performance with regard to the L1b and L2 data products until the end of the mission.





In Sect. 3 the main features of the new L01b data processor are described, as well as the properties of the new data formats
and standards that are applied. In Sect. 4 the changes to electronic calibration are presented, where in Sect. 5 the radiometric
improvements are explained. In Sect. 6 many modifications are presented that are related to the annotation of the L1b data,
especially the flagging algorithms and the geolocation information. The verification approach used for the collection 4 L1b
data is described in Sect. 8. The conclusions are summarized in Sect. 9.

## 2 Changes to the instrument operations baseline

For OMI instrument operations, an orbital scheduling approach is used. Earth radiance measurements are performed on the day
side of the orbit. At the north side of the orbit, near the day-night terminator, the Sun is visible in the instrument's solar port.
Approximately once a day, a solar irradiance measurement is performed. The night side of the orbit is used for calibration and
background measurements. In the following sections all recent changes to the instrument operations are summarized.

### 2.1 Instrument thermal configuration

Degradation of the thermal radiator reduces its ability to remove heat from the instrument. For constant thermal settings, this
leads to a slow heating of the optical bench (OPB) and detectors. The detectors are thermally stabilized by a P-type control
loop with a heater with pulse-width modulation (PWM). From around orbit 70000 onwards, it was observed that the duty cycle
of the PWM of the UV detector heater occasionally dropped to zero, so in fact not stabilizing the temperature at all times
anymore. The best solution to regain control over the detector temperature stability was to reduce the heater power for the OPB
level heaters from 10 W to 8 W. This reduced the thermal load to detectors and radiator and lowered the OBP temperature by
about 1 K and the detector temperatures by about 40 mK. With this change the effective thermal configuration was reset to the
situation the instrument had halfway the mission. This change has no impact on the quality of the calibration because the L01b
processor corrects for thermal dependencies when needed. The duty cycle for the UV detector heater PWM increased from an
average of around 3% to about 13%, which leaves enough headroom until the end of the mission. These new thermal settings
were made active on 26 November (day of year 330) 2019 around orbit 81729.

### 2.2 Changes to the measurement modes

Since the start of OMI routine operations a repetitive scheme of 466 orbits was used to allow a fixed pattern in geolocation
coverage. End of 2019 and during 2020, updates to the nominal baseline were made such that a highly repetitive 360 orbit
baseline was implemented similar to that of TROPOMI. From that point on, the fixed pattern in geolocation coverage was
abandoned and no spatial zoom measurements were performed any more. The instrument calibration measurements were
reviewed and only those calibration measurements were kept in the baseline that had proven useful. Especially all white light
source (WLS) measurements were removed since the failure of this calibration source in orbit 80737. When reviewing the
schedule for all measurements, the future plans for the satellite were also taken into account, such that stable operations is
assured even when Aura will start drifting in LTAN and leave the A-train constellation.



The instrument operation schedule has been updated such that calculation and calibration needed for background correction and random telegraph signal detection can now been done by the collection 4 L01b processor in forward mode without the need for the TMCF system.

## 3   Processor system development

The collection 3 data processing architecture was described in van den Oord et al. (2006). For the TROPOMI L01b proces-

sor a new architecture was used that allows for processing of higher data volumes at higher processing speeds. Also, many lessons learned from the 15 years of experience with the collection 3 OMI processor were incorporated in this TROPOMI design (KNMI, 2017).

### 3.1   Processor architecture

The OMI collection 4 L01b processor is using the exact same architecture as the TROPOMI processor, the major improvements

are:

**data lifetime**  The collection 3 L01b processes measurements one at a time. Once the processing of a measurement is finished, all the data related to that measurement is deleted from memory. Therefore, it is not possible for algorithms to use data from (an)other measurement(s). Also aggregate calculations, such as averaging over multiple measurements, cannot be implemented without complex and cumbersome workarounds. The design of the collection 4 L01b makes it possible to

have dependencies between measurements and perform aggregate calculations.

**processing sequence**  The collection 3 L01b has a single-pass design, whereas the collection 4 L01b has a multi-pass design. A single-pass design results in a program that traverses through all data and the process flow in a single-pass. The multi-pass design is more flexible and allows a program to re-use data during the processing flow. The data that is re-used can be both input data or the (intermediate) results of a processing step. This allows, for example, to initially process background

measurements, and use an aggregate of these processed background measurements in the background correction during the processing of the remaining measurements.

**processing flow configuration**  The processing flow of the collection 3 L01b is table-driven. It uses a table that specifies which algorithms should be executed in what order, and has only a single table per measurement class, which was determined at compile time. The collection 4 L01b also uses tables, but these tables are determined not in code, but in configuration

files. This allows the tables to be loaded at start-up. Another improvement is that the tables allow a more fine-grained processing configuration.

**process threading**  The collection 3 L01b uses a single process / single-threading approach. As a result, the processor will run as fast on a single processor / single core machine as on a multi processor and / or multi core machine. This severely restricts the scalability of the software on modern platforms. For the collection 4 L01b a multi-threading approach was

chosen that allows it to scale better on modern platforms.



**internal data containers** The collection 3 L01b uses generic data containers for storing data. The collection 4 L01b uses the same principle, but with several enhancements. The collection 4 L01b is developed in C++ which has stricter type checking than the C language in which the collection 3 L01b was developed. As the new L01b design allows for dependencies between measurements and for aggregate calculations, the collection 4 L01b data containers can store data for multiple measurements, where the collection 3 L01b would only store data for a single measurement.

**algorithm configuration** The collection 3 L01b uses fine-grained algorithms, which means that whenever possible, a separate algorithm is used for each instrument feature. Additionally, where possible, the algorithms for calculating the correction parameters, applying these correction parameters and any quality assessment (flagging) on the data are separated. The collection 4 L01b reuses this principle and improves it further by allowing algorithms to be plugged in at start-up through the use of shared libraries. Which libraries to load is part of the configuration. This makes it possible to load different versions of a library at start-up, making it possible to easily assess the impact of changes in an algorithm / library.

With the aforementioned enhancements it was now feasible to remove the entire TMCF inflight calibration system, and to incorporate all algorithms that are needed for monitoring, trending and correction for the instrument aging in the collection 4 L01b processor. In addition the same processor can now be used for regular forward processing, near-real time (NRT) processing, as well as reprocessing.

## 3.2 Level 1b data format

The OMI science data is split into 2 detectors (UV and VIS) and 3 bands, with spectral ranges as described in Table 1. The radiance data from detector 1 (band 1 (UV1) and 2 (UV2)) are provided in the OML1BRUG products. The radiance data from detector 2 (band 3 (VIS)) are provided in the OML1BRVG products. Once a month, radiance data is also collected with a higher spatial resolution as given in Table 2. This so-called zoom data is stored in the separate data products OML1BRUZ and OML1BRVZ for the UV and VIS detectors, respectively, and was discontinued in 2020. The solar irradiance data from both detector 1 and 2 (bands 1–3) are provided in the OML1BIRR product. The data from the radiance and irradiance products can be combined to calculate reflectance data.

**Table 1.** Spectral range, resolution and sampling distances of the OMI instrument. Bands 1 and 2 are imaged separately onto detector 1 to allow for different spatial sampling and higher signal to noise in band 1.

| Detector | Band | Total range | Average spectral resolution | Average spectral sampling distance |
|---|---|---|---|---|
| Detector 1 (UV) | Band 1 (UV1) | 264—311 nm | 0.63 nm | 0.33 nm/pixel |
| Detector 1 (UV) | Band 2 (UV2) | 307—383 nm | 0.42 nm | 0.14 nm/pixel |
| Detector 2 (VIS) | Band 3 (VIS) | 349—504 nm | 0.63 nm | 0.21 nm/pixel |

Note that there is spectral overlap between bands 1 and 2 and also between bands 2 and 3.



**Table 2.** Spatial sampling properties in nominal and zoom measurement mode.

| Radiance mode | Nominal | Zoom |
|---|---|---|
| binning factor | 8 | 4 |
| number of pixels UV1 | 30 | 60 |
| number of pixels UV2 and VIS | 60 | 60 |
| nadir pixel size UV1 | 48 km | 24 km |
| nadir pixel size UV2 and VIS | 24 km | 12 km |

Note that the zoom measurements were discontinued in 2020 in line with the modifications to the instrument operations baseline.

The OMI collection 3 L1b products were stored in the HDF-EOS format that was based on HDF-4. These data formats
were abandoned in favor of the TROPOMI standard which uses the NetCDF-4 format (Unidata, 2021), which is based on
HDF-5 (HDFgroup, 2021). The OMI collection 4 products can be read with the NetCDF-4 or HDF-5 libraries, which are
available for a variety of different programming languages. The NetCDF format structures data in groups and datasets. The
group structure is described in detail in Rozemeijer et al. (2021).

The collection 4 products follow a completely different format structure, which is largely based on TROPOMI; the most
notable changes from collection 3 are as follows.

– Radiance and irradiance data are now stored in ascending order of wavelength, which means that for band 1 (UV1) the
spectral dimension is reversed compared to collection 3.

– In the collection 4 L01b processor both the radiance and irradiance data are now corrected for the Earth-Sun-distance,
i.e. normalized to 1 astronomical unit (au).

– The data is no longer split into a separate mantissa and exponent and instead of storing a noise value directly, the noise
is provided as a signal-to-noise ratio on a decibel scale.

– The collection 4 products no longer support re-binning. This means that zoom data will still be generated, but no longer
re-sampled to the global resolution as was done in collection 3.

### 3.3 Level 1b metadata

The baseline for providing metadata for the collection 4 L1b product is governed by the ISO 19115 International Geographic
Metadata Standard (ISO, 2003) together with the ISO 19115-2 extension for imagery and gridded data (ISO, 2009) and Earth
Observation Metadata profile of Observations & Measurements OGC 10-157 (OGC, 2012, 2014). These standards are leading
as prescribed by INSPIRE (JRC, 2010).

In specifying the metadata for the OMI collection 4 L1b products several metadata conventions and standards are taken into
account. Two relevant conventions are related to the use of NetCDF as file format for the L1b products: the NetCDF Climate





and Forecast (CF) Metadata Conventions (CFConventions, 2011) and the Attribute Convention for Data Discovery (ACDD), governed by the Federation of Earth Science Information Partners (ESIP), which is an open networked community.

In addition, two ISO standards are important that are related to the description of collections of Earth Observation (EO) products ISO 19115-2 (ISO, 2009) and to the description of individual EO products ISO 19156 (ISO, 2011), respectively. As
described in the input/output data specification document (Rozemeijer et al., 2021), metadata are included into the NetCDF L1b product as global attributes and as attributes organized into groups-of-groups, based on their intended use. The full metadata specification is described in the metadata specification document (Rozemeijer, 2021).

### 3.4    Degradation correction in level 1b processing

A major improvement to the collection 4 L01b processing method is the introduction of time-dependent calibration key data.
For collection 3 the elaborate TMCF system was needed to facilitate this functionality. For collection 4 the TROPOMI method was used, in which any calibration key data may have a (optional) time-dimension. If this time dimension is present in the CKD file, the processor will generically use interpolation or extrapolation such that optimal calibration key data is generated for each orbit number being processed. The available collection 3 TMCF data was used as input for the analysis to yield the time dependent key data for the collection 4 L01b processor.

### 3.5    Level 1b irradiance processor

An addition to the collection 4 processing system is the introduction of a running-averaged irradiance product. A typical phenomenon with imaging spectrometers like OMI and TROPOMI is the appearance of systematic along-track features in plots of L2 orbital data. This so-called *striping* is mainly caused by random noise and pseudo-random features in the irradiance measurements that become systematic errors in the L2 products. By design, each cross-track position has its own measurement
of the solar irradiance, but apart from spectral sampling and slit function, these may only differ in measurement noise. Thus in L2 retrievals each *cross-track* position uses a slightly different solar spectrum, for all *along-track* positions in the orbit being processed. These subtle spectral differences become apparent as a stripe pattern when visualized.

In order to alleviate this, the collection 3 L2 products used a static irradiance for the whole mission, that was the average over all available 2004 irradiance measurements. The much higher averaged signal to noise in this mean reduced the stripes
significantly, but also introduced another problem on its own. Even-though L2 retrievals were now insensitive to solar port degradation (which was known, but not corrected for), they became sensitive to degradation of the Earth port (which was not confirmed, and also not corrected for). In addition, a static irradiance measurement used over a 17 year mission ignores the subtle changes in the solar output, an effect that could enter the L2 products in the long term.

For collection 4 L2 processing an alternative irradiance product is generated that consists of the running average over 100
daily irradiance measurements, yielding an improvement of the signal-to-noise ratio with a factor of 10. Because it is a running average it still captures the subtle changes in solar output, and due to the degradation corrections now in place for the Solar and Earth ports, no instrumental effects will enter the L2 products.





## 4 Improvements to the electronic calibration

Two CCD detectors are used in the focal plane area to detect the incoming radiation. These detectors are capable of on-chip
binning, after which the signal is amplified and digitized in the electronics and logical unit (ELU) subsystem. For all details,
the reader is referred to the OMI collection 4 ATBD (Ludewig et al., 2021), added as supplemental information to this paper.
In the following sections all changes with regard to the electronic calibration are described that occurred between collection 3
and collection 4.

### 4.1 Analog-to-digital conversion

Due to a historic decision, in the collection 3 L01b processor as well as throughout the entire on-ground calibration campaign,
an erroneous value of 4096 DN / 5.0 V = 819.2 DN/V was used as the analogue to digital conversion (ADC) factor. As this
factor was used consistently, the error was compensated by the calibrated voltage-to-charge conversion factor further on in the
algorithm chain.

For the collection 4 L01b processor, the ADC conversion factor is corrected to the conversion factor that was established
during ground testing of the detection sub-system. The voltage-to-charge conversion factor is adjusted with the inverse of the
change to the ADC conversion factor, so that the overall calibration is not changed.

The electronic chains for the detector consist of a number of ELU components that, combined, give rise to an (additive)
electronic offset and a (multiplicative) electronic gain for which the L01b processor has to correct the observed signal. The
electronic *offset* consists of various contributions, and is corrected for as part of the analog-to-digital conversion. The method
for offset correction is similar between collection 3 to collection 4: for each of potential gain settings the offset is calculated
using a read-out register (ROR) part, a gain-dependent part and a gain-independent part. The electronic *gain* correction is
described in the next section.

### 4.2 Amplifier electronic gain ratio

Concerning the gain correction in the L01b processor, four gain settings can be selected when using the instrument. One of
the four is the so-called neutral gain, equivalent to a electronic amplification with one. The *relative* gain is defined by the ratio
between the amplification of the gain setting of the measurement and the amplification of the neutral gain setting.

In collection 3 a static value for each of the 4 gain ratios were used that were calibrated using year 2005 in-flight measure-
ments. The collection 4 a temporal axis was included because the gain ratios are now known to drift in time. For all orbits that
contain a gain ratio calibration measurements, the four gain ratios are computed and hence form an entry in a look-up table. The
L01b processor linearly interpolates between the orbits in the table to obtain the gain ratios for the orbit under consideration.
As can be seen in Fig. 1, the collection 4 gain ratios address a 0.4% drift in the instrument gain ratios that was not accounted
for in collection 3.





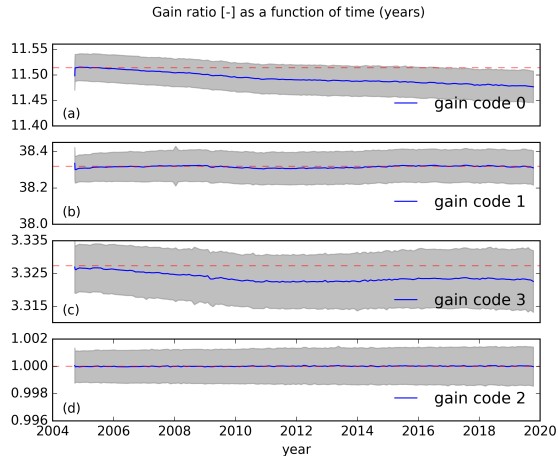

**Figure 1.** The trend over 15 years for the four gain ratios of the UV detector. Results are similar for the VIS channel and are not shown here. The panels show the calibrated gain ratio in each month as a blue line, together with the $1\sigma$ standard deviation indicated by the gray shaded area. The red dashed line shows the gain ratio values at the start of the mission.

### 4.3 Register full well

After inspection of some extreme Sun glint events with a large number of saturated pixels, it was found that the register full
well saturation was reached and hit a ceiling at signal values lower than the collection 3 calibration key data (CKD) limits.
Therefore, an update to these limits was made by analyzing in-flight calibration LED measurements with large exposure times
and large binning factors, where the pixels were surely saturated because of register full well limits. The collection 4 register
full well values are now set to $2350000 \text{ e}^-$ and $2300000 \text{ e}^-$ for the UV and VIS detectors, respectively (these were both set to
$2500000 \text{ e}^-$ in collection 3). The register full well limit factor is left unchanged at 0.95. Also note that the values for the *pixel
full well* did not change between collection 3 and collection 4.

### 4.4 Detector non-linearity

The charge-to-voltage conversion consist of two steps. The first, as mentioned before in Sect. 4.1, is a straightforward unit con-
version, the second is the correction for the CCD non-linearity. The non-linearity stems from the charge-to-voltage conversion
at each CCD output node, and is, by construction, the same for all pixels. In principle, longer exposure to a signal leads to more
charge being built up in the register of a pixel of the CCD. To lowest order, this shows a linear behavior; deviations from this
linear behavior are captured in the non-linearity calibration key data.

For collection 3, a smooth polynomial as a function of the signal has been determined from in-flight LED measurements
using a sequence of 15 exposure times. For collection 4, the method for derivation was improved using the TROPOMI ap-
proach (KNMI, 2017). The collection 3 curve more or less intersects the point ($1 \text{ Me}^-$, $20 \text{ ke}^-$). This feature is now imposed



as an algorithm constraint in the collection 4 method. Differences with the collection 3 CKD will therefore mostly occur on both sides of this anchor point.

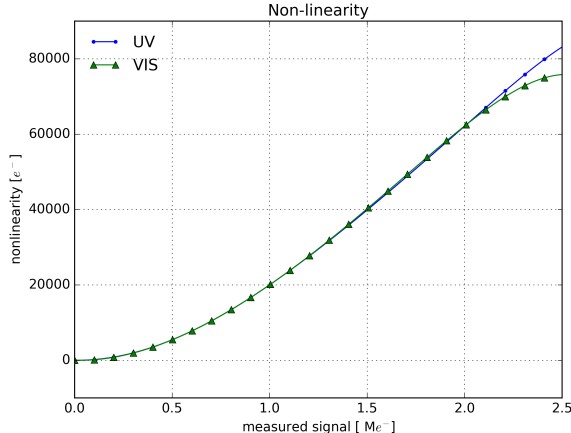

**Figure 2.** The absolute non-linearity curves for both the UV detector and the VIS detector. The VIS curve is less accurate for higher signals due to the lack of images in these higher signal ranges. At lower values it overlaps with the UV curve. Just as in collection 3, the curve for the UV detector is therefore used for both UV and VIS.

The non-linearity in collection 4 is expressed in Chebyshev polynomials. Since the curve obtained from the VIS detector strongly resembles the UV curve (see Fig. 2), at least in the signal range available for VIS, the CKD for the VIS detector has been copied from the UV detector CKD, just as in the collection 3 CKD.

### 310 4.5 Detector pixel quality flags

For collection 4, a detector pixel quality flags (DPQF) map similar to TROPOMI is created. In collection 3, the attribution of pixel quality is done by computing three (out of 31) flags from certain calibration measurements. These three flags are:

**deadWLSlow** In a WLS measurement, pixels that have a too low signal value compared to the immediately neighboring pixels are detected.

**deadDISC** So-called disconnected pixels: a small set of pixels that during on-ground calibration measurements showed off-nominal signal. This set only consists of nine pixels in the non-illuminated area between UV1 and UV2 on the UV detector.

**bad/dead DChigh** A dark current in a pixel that is too high compared to a fixed threshold.

The DChigh category poses a problem, since the general increase of the dark current during the mission leads to a growing 320 fraction of flagged pixels for this criterion, reaching 20% in 2019. This fraction makes the flags no longer useful. Instead of adjusting the criterion for the category of high dark-current pixels, this partial flag was entirely discarded. The justification is





that a high dark current in itself is not a problem, as long as it is correctly negated by the background correction algorithm in the L01b processor. This shifts the problem to the question if the dark current can be adequately corrected. The answer is affirmative, as long as it is stable during a reasonably short time interval, i.e. during the time interval in which both the illuminated frame and dark images that constitute the background image are measured. The assessment of this stability of dark current is done in the RTS flagging algorithm described in Sect. 4.6.

For the DPQF map construction for collection 4 the 'dark current' and 'disconnected' criterion were removed. The WLS criterion was fine-tuned, and all years from 2005 to 2019 were inspected, and a stationary DPQF map based on a majority vote was established. This map is valid for the past 17 years, but can extended when new dead pixels may occur in the future.

For this fine-tuned approach the monthly unbinned WLS measurements were independently reprocessed. For each measurement, three frames in time are available. By taking the median (of these three), per pixel, transient measurements are effectively discarded. Further, a median filter in the 3×3 region around each pixel is used to detect pixels with deviating signal, exactly as in collection 3. More precisely, a pixel is flagged if its signal value divided with the median value is too far from unity as demonstrated in Fig. 3. The resulting static map contains very few flagged pixels: 9, 16 and 15 pixels for the UV1, UV2 and VIS channels, respectively.

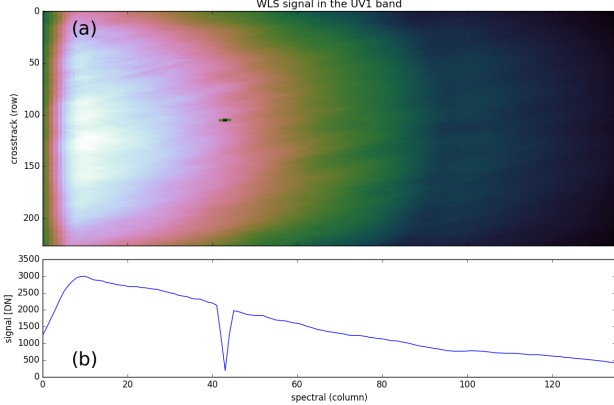

**Figure 3.** WLS signal in the UV1 band, corrected for background. Panel (a) shows the signal in the whole UV1 region, with a distinct dark spot around column 40 and row 100. Panel (b) shows the cross-section of the signal through the structure.

## 4.6 Random telegraph signal

Random telegraph signal (RTS) is the phenomenon that detector pixel dark current changes between discrete values on random timescales, and becomes a problem if it occurs on timescales shorter than the period over which the background radiance measurements are averaged. In collection 3, the RTS map was derived from a time series of 30 consecutive days after an elaborated analysis involving statistical measures like mean, standard deviation, skewness and kurtosis, using the TMCF system. For collection 4, the RTS concept is revisited, combined with a re-assessment of the pixel quality map (see Sect. 4.5).



Detector pixel dark current has increased considerably during the mission, both on average and for many individual pixels. On closer inspection, almost all detector pixels have been at least hit once by cosmic particles during the mission, resulting in a higher dark current for a wide range of time intervals. Dark current itself is however no longer considered a major problem.

Now, given that all radiance measurements are always corrected with background correction, the only criterion should be that the dark current (that forms the main part of the background image) in both radiance and its associated background should be the same. More precisely, a pixel in the averaged background image that is constructed from up to 15 consecutive orbits should not show any RTS *on that timescale*.

Therefore, a comparison between the expected noise (part of the L01b product, and consisting of the sum of read-out noise

and shot noise) and observed noise (the temporal variance of the ca. 800 dark frames accumulated in a day) of a pixel is sufficient to determine if a pixel suffers from RTS on this timescale. Note that, in order to correctly compute the observed noise, transient pixels have to be filtered out first, as discussed in Sect. 6.5. The collection 4 L01b processor creates the RTS map of binned pixels (according to the radiance binning schemes) together with the background radiance products, for every orbit, based on the data collected in the previous 15 orbits.

## 5 Improvements to the radiometric calibration

In the following sections all changes with regard to the radiometric calibration are described that occurred between collection 3 and collection 4.

### 5.1 Instrument radiometric calibration

During the on-ground calibration period, the radiometry of the instrument was determined, and is referred to as the so-called

day-one calibration. It is noteworthy that the instrument bi-directional scattering distribution function (BSDF) was measured using a single calibration source, sequentially observed through the Sun and Earth port of the instrument.

However, the L10b data processor generates separate products for the two ports, both of which need to have absolute radiometric values attached to the provided radiance and irradiance values. Therefore, the Sun port sensitivity was measured on-ground using a national institute of standards and technology (NIST) traceable light source. This sensitivity calibration is

used by the L01b processor to generate the irradiance values of the Sun observations. This instrument irradiance sensitivity is combined with the aforementioned BSDF calibration to yield the instrument radiance sensitivity of the Earth port, which in turn is used by the L01b processor to calculate radiance values for the Earth observations.

After careful consideration it was decided not to change this radiometric calibration, and to keep the day-one calibration because no potential improvement was found for collection 4. A small change however is that in collection 3 the sensitivity

calibration, as used by the L01b data processor, was provided as a function of wavelength in the calibration key data. For collection 4 the TROPOMI convention was used, and the calibration key data was converted to be a function of detector pixel.

In addition, some minor cosmetic corrections were made in the CCD areas outside the science region. In the collection 3 calibration values were encountered that were large, yet not fill values. To make sure that any kind of interpolation algorithm





applied on these data would not have to accommodate to these values, the regions were filled with the values in the closest

row or column inside the science region. Inside the science region, values are now clipped in collection 4: values that are much higher than is to be expected are reduced to fall within a reasonable range. This was done pragmatically, using a separate maximum for the low and high wavelength part of the detector region, and different values for radiance and irradiance, as given in Table 3.

**Table 3.** Thresholds for clipping radiance and irradiance (given in their corresponding units) in the left and right parts of the three bands.

| region | UV1 | UV2 | VIS |
|---|---|---|---|
| radiance left | $2.0 \cdot 10^{-12}$ | $1.0 \cdot 10^{-11}$ | $1.2 \cdot 10^{-11}$ |
| radiance right | $2.0 \cdot 10^{-13}$ | $8.0 \cdot 10^{-11}$ | $5.8 \cdot 10^{-12}$ |
| irradiance left | $2.0 \cdot 10^{-10}$ | $5.0 \cdot 10^{-9}$ | $1.2 \cdot 10^{-9}$ |
| irradiance right | $2.0 \cdot 10^{-11}$ | $9.0 \cdot 10^{-9}$ | $3.4 \cdot 10^{-10}$ |

## 5.2 Instrument radiometric degradation

Over the course of the mission, the instrument performance may change due to a variety of effects. Detector or electronic properties may degrade due to radiation damage by cosmic particles, and the throughput of the instrument may degrade due to contamination of optical surfaces. Thus, while the initial absolute calibration remains unchanged, the temporal degradation of the instrument must be corrected for in the irradiance and radiance sensitivity data.

It should be noted that when the reflectance is calculated by division of the radiance by the irradiance some degrading

effects that are common to both the radiance and irradiance port cancel out. Effects that are not common to both ports, do not cancel out, and because this is detrimental to the quality of the L1b product these effects should be corrected. By separately characterizing the *full* change in radiance and irradiance, the corrections will ensure that the changes to the instrument bi-directional scattering distribution function due to aging are compensated such that the calculated Earth's reflectance is not affected throughout the mission.

The two instrument configurations used for Solar and Earth measurements are distinctly different. In order to understand and correct the observed instrument degradation these differences need to be taken into account. Earth observations are performed using the telescope that consist of a primary and secondary mirror in the light path towards the spectrometers. The Sun is observed over one of the three available diffusers, using a folding mirror to direct the light to the spectrometers. This folding mirror bypasses the telescope primary mirror, which is thus not included in the Solar measurements. The QVD diffuser is the

main diffuser used for the daily Sun observation and yields the L1b irradiance product. The ALU1 regular diffuser is used once a week, and the ALU2 backup diffuser once a month; these observations are not publicly reported, but are stored in the L1b calibration product.

Combining Sun measurements using the three different diffusers we can make a first-order guess as to the origin and spectral shape of the observed degradation. In Table 4 the degradation fractions in 2017 relative to the first measurement in 2005 are





**Table 4.** Wavelength averaged degradation over the period 2005–2017, derived from Sun observations over the three diffusers, for all three channels. In the top three rows the signal fractions in 2017 relative to the first measurement in 2005 are given, the lower three rows list the contributions of the elements in the QVD path.

| year 2017 | UV1 | UV2 | VIS |
|---|---|---|---|
| ALU 2 monthly | 0.97 | 0.975 | 0.975 |
| ALU1 regular | 0.96 | 0.972 | 0.974 |
| QVD daily | 0.915 | 0.95 | 0.963 |
| QVD path | 8.5% | 5.0% | 3.7% |
| QVD diffuser only | 5.5% | 2.5% | 1.2% |
| other sources | 3.0% | 2.5% | 2.5% |

given. The numbers are calculated as the average over all wavelengths within each of the three channels. Under the assumption that the diffuser degradation is exposure based, it is expected that the backup ALU2 diffuser has the lowest degradation because it is only used once a month. The degradation of regular and backup diffusers is not the same, and differs per their exposure ratio From this follows that the ALU1 or ALU2 diffuser plates did not degrade significantly themselves, but rather a common component downstream. The most likely source is the folding mirror, that then would account for nearly half of the total

degradation observed in the QVD path. In addition, as the ALU2 values are comparable for all channels, the downstream degradation has no strong wavelength dependency. This in turn suggests that no strong wavelength dependent degradation has occurred in the radiance port (ignoring the unknown telescope primary mirror degradation).

In order to study the wavelength dependence in more detail the total degradation as observed in the QVD over the period 2005–2021 is given as a function of wavelength in Fig. 4. Also shown is the degradation of the QVD when divided by the

ALU2 diffuser, to remove the common degradation and isolate the QVD optical components. These observations support that most of the wavelength dependent degradation occurs in the QVD, and not in the other diffusers. Because there are more unknowns than measurements it is not feasible to identify the exact degradation of each component separately. The folding mirror and spectrometer degradations cannot be quantified independently without additional information obtained from Earth radiance measurements.

We therefore have adopted a pragmatic way for the radiometric degradation correction in collection 4 L1b data: for both ports we use independent methods to estimate the total observed degradation, and correct for these. Common degradation in the spectrometers is then included in both corrections, and cancel out in the calculated reflectances. The degradation model chosen assumes approximately 3–4% row dependent, but wavelength independent degradation for the Earth port. Furthermore, the model assumes that all remaining (small) wavelength dependent degradation can be attributed to the folding mirror.

The degradation of the Sun port and Earth port are analyzed and corrected as described in Sect.s 5.4 and 5.5, respectively. Because the correction for the Sun port depends on an accurate correction of the dependence of the observed irradiance on the solar incident angle on the diffuser, this topic is treated first in Sect. 5.3.



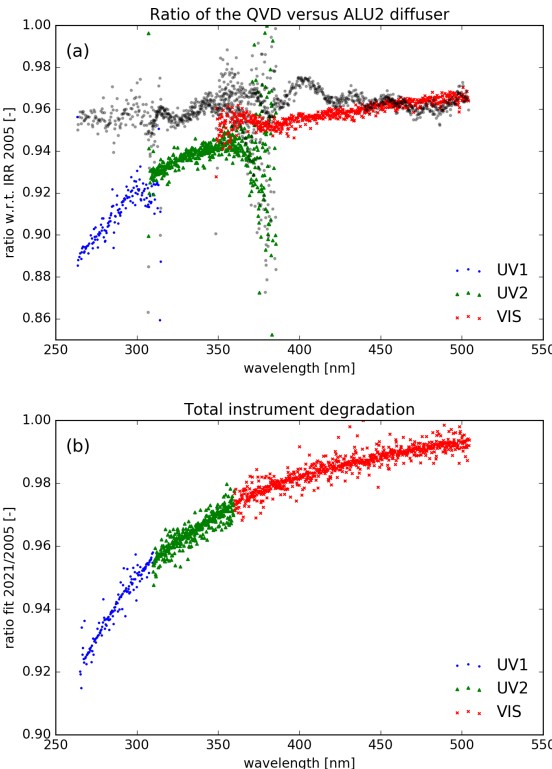

**Figure 4.** Total instrument degradation observed through the QVD over the period 2005–2021 is shown in the top panel for all wavelengths. The UV1, UV2 and VIS data is shown in blue dots, green triangles and red crosses, respectively. In grey circles the total instrument degradation through the regular ALU1 diffuser is shown, which is a representative measure of the FM change and spectrometer change. The larger spread is caused by diffuser speckle that is more prominent in the aluminium diffusers than in the QVD. Clearly there is an overall 4% degradation with no strong wavelength dependence; for the ALU2 diffuser this dependence is even lower. The lower panel shows the ratio between the QVD and ALU2 backup diffuser. This isolates the QVD components from the common components in the optical path, and clearly shows the wavelength dependent degradation of the QVD.

## 5.3 Relative irradiance

The relative irradiance describes the dependence of the observed irradiance on the solar incident angle on the diffuser. During each daily measurement of the irradiance, the elevation angle varies approximately from -4° to +4° degrees due to the movement of the satellite, but the solar azimuth angle remains more or less constant. During the year this azimuth angles changes with the season over a range of +18° to +32°. The relative irradiance correction in the L01b processor removes this angular dependency and guarantees that the L1b irradiance product can be generated from any daily measurement regardless the used observational angles.





The collection 3 calibration key data were determined using irradiance data acquired in the year 2005. The dependence on azimuth angle and elevation angle for collection 3 were parametrized using high order (12 and more) polynomials. The collection 3 CKD consist of the fit coefficients for each detector row (across-track position) and 11 azimuth bins and 10 elevation bins.

For the determination of the collection 4 CKD a completely new method was used. Due to the fact that in the first 5 years
of the mission the azimuth range covered within one year differed quite a bit, it was decided not to base the correction on one particular year of data but to use the entire period 2005—2020 data. Note that the irradiance data used in the analysis are corrected for degradation, in first order, using a reference method described below, which is not the more thorough correction described in Sect. 5.4.

The collection 4 CKD consists of the actual correction factors as function of the azimuth angle (equidistant grid with 280
grid points), elevation angle (equidistant grid with 200 grid points), all across-track position and wavelength windows (these windows are described below). The analysis is only performed for the quartz volume diffuser (QVD) since this is the diffuser that is used for the L1b irradiance product. For the aluminum 1 (ALU1) and aluminum 2 (ALU2) diffusers no new analysis is performed because these are only used for calibration and monitoring purposes. Instead, the collection 3 calibration is re-used for these two difusers, and the corresponding polynomial CKD is converted directly to the collection 4 format.

The full wavelength spectrum can be sensitive to small wavelength shifts and thus cause problems when applying the correction to different years. Therefore, the irradiance data is reduced in the wavelength dimension to wavelength windows. A window size of 10 nm is used in which the data is averaged using a triangular weighting function, with weight 0 at the edges of the window and weight 1 in the middle of the window. For VIS this leads to 12 wavelength windows, for UV2 to 5 wavelength windows and for UV1 to 3 wavelength windows.

The irradiance data of 2005–2020 are combined, and the irradiance reference observation is determined. This is the average irradiance spectrum that corresponds to the reference angle (azimuth angle 23.75° and elevation angle 0.0°). The reference sample is an average of all values in a window of 0.3° around the reference angle. All data is now normalized with respect to this reference spectrum, which effectively is a first-order degradation correction. Note that this degradation correction is not the same as the one used for Solar port degradation correction as described in Sect. 5.4 because that correction relies on the
availability of this relative irradiance correction.

The time–measurement dimension is split into a day-dimension and an elevation-dimension, resulting in a 4 dimensional irradiance data cube. An equidistant grid for the elevation angle is defined in the range -4° to +4° with a step size of 0.5°. This leads to 200 grid points, onto which the irradiance is re-gridded using interpolation.

The irradiance data is sorted by increasing value of the azimuth angle. An equidistant grid for the azimuth angle is defined
in the range 18° to 32° with a step size of 0.5°. This leads to 280 grid points, onto which the irradiance is re-gridded in the day dimension. Subsequently the irradiance is smoothed in both the azimuth dimension and the elevation dimension. The resulting data cube is the calibration key data to be used by the L01b processor. Some cross-sections are shown in Fig. 5 for the VIS channel to exemplify the smoothness in all direction, as would be expected from the optical properties of the diffuser.





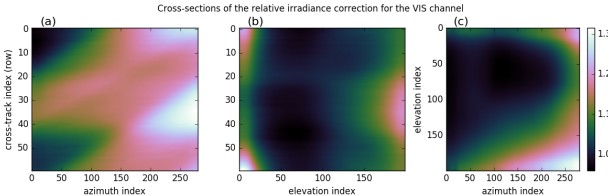

**Figure 5.** Cross-sections of the relative irradiance correction for the VIS channel. Results are similar for the UV1 and UV2 channels and are not shown here. The panels show the across-track position versus azimuth (a), across-track position versus elevation (b), and elevation versus azimuth for a single wavelength (c).

When the CKD is used in the collection 4 L01b processor the irradiance does no longer show dependence on the azimuth angle or elevation angle. This can be seen in Fig. 6 where for the VIS channel the irradiance data is plotted before and after correction, for all prevalent elevation angles within the measurement and for a single azimuth angle of $18.32°$. For the other channels and other azimuth angles the results are similar and not shown here. As can be seen, the correction effectively reduces the dependence on the azimuth and elevation angles.

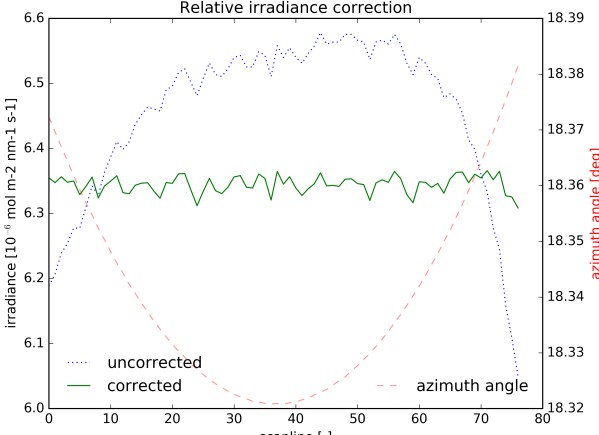

**Figure 6.** Irradiance data for the VIS detector before (blue dotted line) and after the relative irradiance correction (green line) by the L01b processor. Data is shown for a single orbit, so more or less constant solar azimuth angle around $18.32°$ (red dashed line). Within the measurement the solar elevation angle changes from $-4°$ to $+4°$ in the time dimension.

## 5.4 Irradiance degradation correction

In order to be able to calculate the Earth reflectance, the Sun is observed on a daily basis over the primary QVD diffuser as described in Sect. 5.3. This measurement is done using a folding mirror that is unique to the solar port optical path, and does not include the telescope primary mirror that is used in the radiance optical path (Dobber et al., 2006). However, both the





diffuser and folding mirror can degrade in throughput due to photo-polymerization of surface contaminants. A correction of this potential degradation is needed as the effect would otherwise enter the calculated Earth reflectance. The assumptions on which the degradation correction is performed are threefold:

1. There is no optical degradation at the start of the mission.

2. Based on the observed instrument changes optical degradation appears related to solar exposure, which accumulates uniformly over time.

3. The output of the Sun varies less over the mission time period than the uncertainty in optical degradation over the same time. Therefore, using a constant Sun yields the most accurate measure of irradiance sensitivity change with time.

With these assumptions the general approach to determine the correction is to compare a selection of observed solar measurements during the mission with the first observation. By dividing each measurement with the first (reference) solar measurement, all observations are normalized to this one, and the reference measurement becomes unity. These normalized observations are not used directly, but are filtered to yield smooth spectral and temporal degradation curves, as described below. Onwards, the L01b processor uses these numbers inversely to correct for the observed degradation in the solar port in a way that removes the anticipated smooth degradation, but still retains the the fine-spectral and temporal details in the measurements.

The daily QVD measurements that are used in this derivation are all measured at a single specific day in each year (October $1^{st}$). Therefore, the azimuth angle of these measurements is relative constant between 28.4° and 27.8° over the entire mission. These daily observations contain 84 measurements with changing elevation angles, and this angular dependency on the incident elevation angle is addressed by the relative irradiance correction as described in Sect. 5.3. The average of these 84 measurement has sufficient signal to noise, and also reduces the effect of diffuser speckle due to white light interference.

The resulting 17 solar measurements normalized to Day 1, one for each year of the mission, retain their across-track dependency because observed variations in the rate of change are significant. For each measurement and row, the degradation spectrum per channel is smoothed in the wavelength direction because we do not expect sharp features in any optical degradation. The obtained irradiance degradation correction is shown in Fig. 7 for mission year 17. Clearly the degradation is wavelength dependent, where the UV1 channel has the strongest dependency towards the shorter wavelengths. The asymptotic change at longer wavelengths does not appear to be unity. This suggests that 2% – 3% of the observed change is independent of wavelength and not a result of optical degradation. It is also evident that the degradation can be strongly row dependent, especially for the UV1 channel.

In Fig. 8 the time dependency of the observed degradation at three wavelengths per channel is shown for the entire mission. It can be seen that for shorter wavelengths in the UV, the degradation can be as high as 20%, while for the long visible wavelengths the degradation is less than 5%. This correction in the L01b processor addresses the total degradation of the Sun port, so all contributions from the QVD through the detector are corrected, including the folding mirror. The individual contribution of each of these components are not known, but are not relevant for the quality of the resulting L1b irradiance data product.



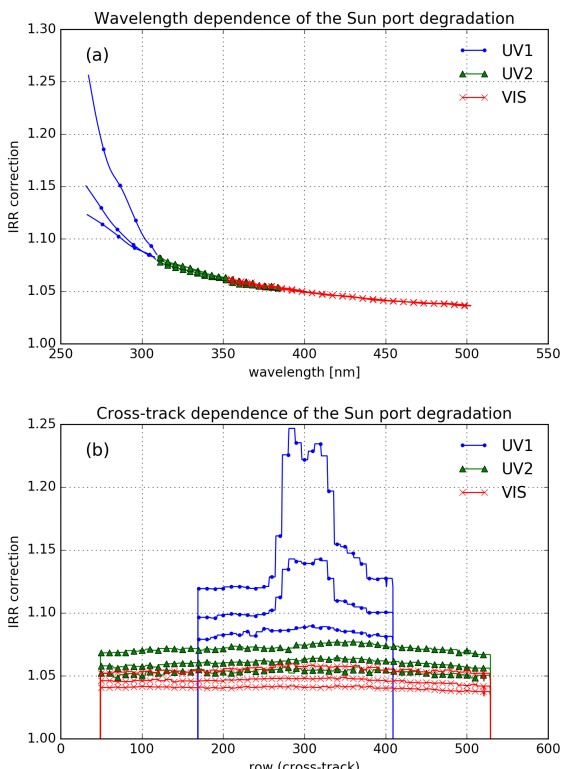

**Figure 7.** Observed degradation of the Sun port for mission year 17. The UV1, UV2 and VIS channels are plotted in blue dots, green triangles and red crosses, respectively. The upper panel shows the wavelength dependence for three rows, the lower panel the row dependence for three selected wavelengths in each channel.

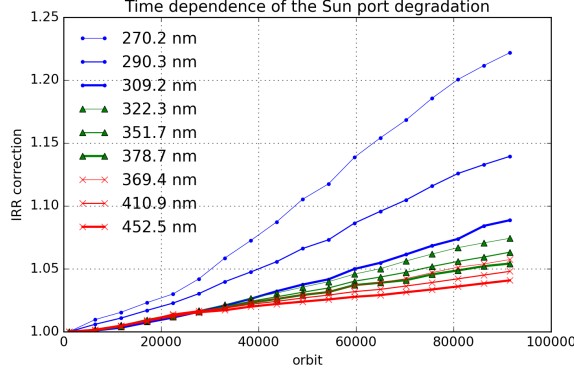

**Figure 8.** Time dependence of the degradation of the Sun port as observed over the entire mission, for three selected wavelengths per channel for detector row 300.



## 5.5 Radiance degradation correction

As described in Sect. 5.2 the OMI on-board calibration system does not support a direct determination of sensor changes affecting Earth backscattered radiance measurement. While the instrument design does incorporate multiple solar diffusers to help isolate the diffuser degradation, a folding mirror is present in solar irradiance measurements that is not present in Earth-view measurements. A portion of the irradiance change indicated by Fig. 4, 7 and 8 is likely a result of degradation in this folding mirror's reflectivity, a degradation that does not affect Earth radiance measurements. Furthermore, the primary telescope mirror is bypassed for solar measurements, so any degradation of its reflectivity will go undetected in the solar calibration measurements.

An estimation of the instrument changes affecting Earth radiance measurements can be obtained using scene-based techniques. Such techniques have been previously used for instruments lacking adequate on-board calibration systems (Wellemeyer et al., 1996). These techniques tend to work well at wavelengths where atmospheric absorption is small and most of the observed radiance change can be attributed to the instrument. At other wavelengths, especially shorter than 330 nm where ozone absorption is significant, variations in the absorption cross section with wavelength can help to constrain the wavelength dependence of the instrument degradation (Herman et al., 1991).

The technique chosen to track OMI calibration changes is to monitor top of the atmosphere (TOA) reflectances over Antarctica and Greenland (Jaross and Krueger, 1993). Permanent snow cover over these continents represent stable surface reflectivities at levels well below the observed instrument changes. Ice radiances were previously used to adjust the BSDF calibration of OMI at the beginning of the mission (Dobber et al., 2008a). Snow-covered ice reflectivity has a complicated, poorly known directional character, which is the primary source of uncertainty when validating measured TOA reflectances. This directionality can also alias into apparent instrument response change as viewing conditions drift, but the stable Aura orbit means that the OMI's view angles are highly repeatable and knowledge of the directional reflectivity is less important.

Figure 9 contains images of measured TOA reflectances over Antarctica. Reflectance is calculated by normalizing the measured radiances with a fixed solar measurement from the end of 2004, so the results are indicative of Earth radiance changes alone. The figures show the ratio of monthly mean OMI reflectance measurements for the UV2 and VIS bands over the Antarctic ice sheet in January of 2005 and 2019. The ratios are plotted as a function of cross-track position and wavelength and were smoothed with a 1 nm boxcar filter. Band 2 (UV2) spectra below 335 nm and above 360 nm are excluded from the analysis to avoid regions affected by ozone absorption and poor signal respectively. The results show around 2% degradation in the radiance channel on both detectors over roughly 1.5 decades. The change is mostly wavelength independent with a cross-track dependence of approximately 1%. The localized pattern of additional spectral and cross-track dependence on band 3 (VIS) between 350 and 385 nm is identified as the spectral region affected by the dichroic. Banding in the spectral dimension is evident at the Ca Fraunhofer lines at 390–400 nm and the G-band near 430 nm on band 3 (VIS), and at other solar lines between 340–360 nm on band 2 (UV2).

The data indicate that the instrument response has decreased by 2%–3% since launch in most non-anomaly rows and at all wavelengths appropriate for the technique. Changes less than 2% can be seen in the first 10 rows, but is not certain if





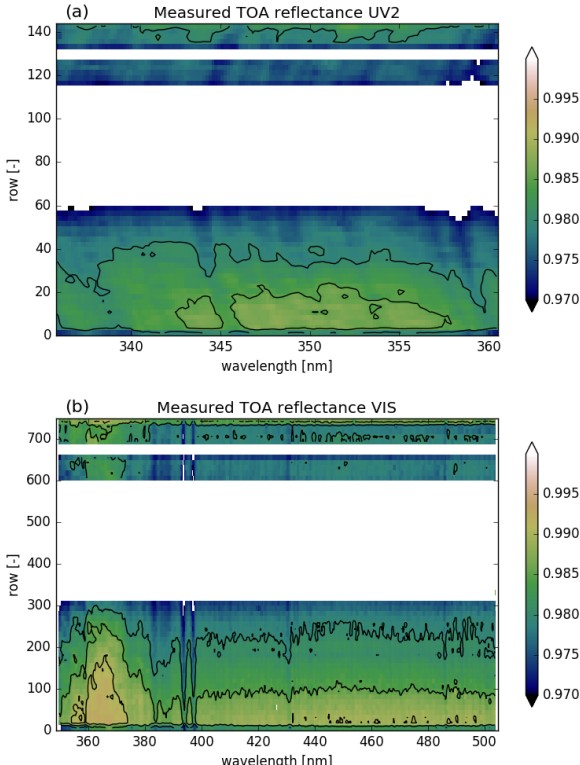

**Figure 9.** Averaged monthly measurements of TOA reflectances over Antarctica in January 2019 normalized to the same month in 2005 for the UV2 (panel (a)) and VIS (panel (b)) channels. Detector rows affected by the instrument row anomaly have been eliminated.

this represents an actual row-dependent sensor change or an artifact of the analysis technique. Comparisons with independent

techniques (Dobber, 2008a) indicated larger uncertainties near the swath edge. Apart from the dichroic region there is little sign of enhanced decreases at short wavelength that are characteristic of optical degradation. The curious behavior in the dichroic region is also observed in the solar data (see Fig. 4), and is discussed in more detail in Sect. 8.

The time series of Antarctic 340/500 nm signal ratios shown in Fig. 10 supports the observation of wavelength-independent change. It also leads to a hypothesis that instrument change affecting Earth radiances is primarily a result of electronic rather

than optical degradation, the row anomaly notwithstanding. The asymptotic solar irradiance change of 3% seen in Fig. 7 is consistent with this hypothesis. In the last four or five years there is very slight drift in the 340/500 nm ratio that is not present earlier in the record where things are very flat. This effect is so small that the conclusion of wavelength-independent sensitivity change is still valid within the uncertainty of the analysis.

The ensemble of Antarctic radiance measurements contain no obvious step changes nor anything more complicated than

a linear dependence on time. The radiance corrections used in the collection 4 processing similarly assume a linear time dependence in all rows. The linear fit has a standard deviation of < 0.25% for rows 1–20 and < 0.5% for all rows. This linear





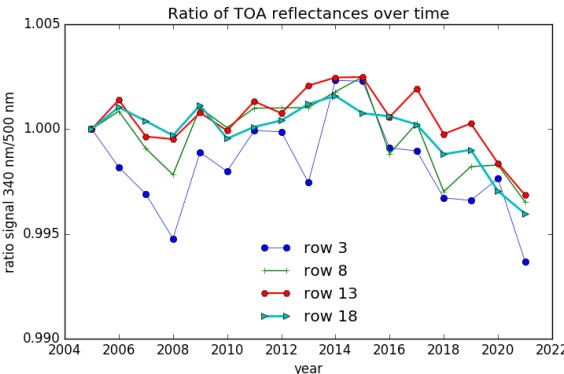

**Figure 10.** Time series of the ratio of measured TOA reflectance at 340 nm to 500 nm, in UV2 and VIS, respectively.

time dependence means that the correction can be easily extrapolated and used for future processing. Fig. 11 contains the expected change as of orbit 100000 as a function of detector row. If necessary, the calibration will be updated as new data are obtained. No attempt has been made to compensate for the RA changes in affected rows, so these rows remain unusable. The

same correction factor is used at all wavelengths. There is very likely some spectral dependence in the Earth radiance response, but the ice data do not provide evidence for such changes.

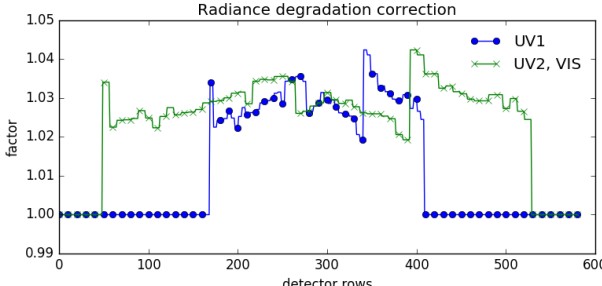

**Figure 11.** Radiance degradation correction factor (multiplicative) for orbit 100000 as a function of the detector rows for the UV1, UV2 and VIS channels.

## 6 Improvements to the annotation data

### 6.1 Spectral calibration

The spectral calibration algorithm from collection 3 has been changed to a monitoring algorithm in collection 4. Where the

objective for collection 3 was to provide a *calibrated* wavelength annotation in the L1b radiance and irradiance products, for collection 4, the purpose is now to monitor the (semi static) wavelength annotation as provided in the CKD. The results of this





monitoring algorithm are only stored in the L1b calibration products, and no longer provided in the radiance and irradiance products. In the collection 4 L1b radiance and irradiance products only provide the wavelength *annotation*, as described in Sect. 6.2.

The spectral calibration algorithm from collection 3 used a two-step approach. In the first step, for a set of narrow spectral windows in each band, a spectral shift was calculated, relative to the wavelength annotation from the CKD. The spectral shift was calculated by fitting reference spectra to the observation data, using an iterative, non-linear fitting method. In the second step, based on the results from the first step, for each band a new wavelength polynomial was derived, describing the relation between the detector pixels and the wavelengths. This *calibrated* wavelength polynomial was intended as an alternative to the 570 wavelength *annotation* to be used in the L2 algorithms. However, the accuracy of the calibration at level 1 is less than what can be achieved at level 2, and therefore this approach was abandoned in collection 4.

The wavelength monitoring algorithm for collection 4 is based on the first step of the spectral calibration algorithm from collection 3. The results of the spectral fitting of the narrow windows are written to the L1b calibration product.

## 6.2 Wavelength annotation calibration

The collection 3 wavelength annotation CKD was based on a polynomial for each row with 5 coefficients w.r.t. column number for the nominal wavelength map. Furthermore, wavelength shifts were applied to all these coefficients due to optical bench (OPB) temperature changes and inhomogeneous slit illumination (e.g. scene changes in the flight direction due to cloud edges).

The order of these polynomials is higher than can be expected physically, and results in numerically unstable behavior near the edges of the bands. Furthermore, this over-fitting can result in unexpected behavior for extreme values of other 580 input variables like the OPB temperature as well. Therefore, the wavelength annotation has been re-calibrated using in-flight irradiance and radiance measurements. The theory for this topic is treated in the OMI ATBD (Ludewig et al., 2021).

### 6.2.1 Nominal wavelength map

Firstly, the collection 4 nominal wavelength map has been established. Using unbinned irradiance measurements, the average of all irradiance measurements during orbit 1157 is used for this analysis. The analysis has been repeated using other nearby 585 orbits of this type with similar results. Using the nominal wavelength map of the collection 3 processor, the observed irradiance spectrum is first corrected for the Doppler shift and is then fitted to a high-resolution solar reference spectrum (Dobber et al., 2008b) that has been convolved with the OMI slit function. This fit is done for a number of spectral fit windows.

In Fig. 12 the observed irradiance spectrum is shown for all rows, along with the solar reference spectrum for the UV1 channel. The measured irradiance spectra can be seen to follow the reference quite well already, while slightly diverging at the 590 edges of the bands. Care has been taken to select the spectral fit windows such that these diverging regions are not included. For each of the spectral windows, a root finding Levenberg–Marquardt optimization method is used to find the fit parameters wavelength shift, intensity, background and slope. After an iterative process, the resulting set of optimal fit parameter values, when applied to the irradiance measurements, follows the reference spectrum as closely as possible, with minimal residuals.





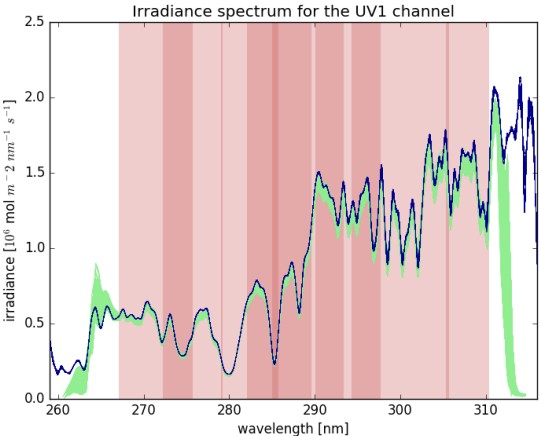

**Figure 12.** Measured UV1 irradiance spectrum in orbit 1157 plotted in shades of green for all rows, versus the collection 3 nominal wavelength map. The high-resolution solar reference spectrum is depicted in blue. The spectral fit windows are shown as red vertical bands, with some overlap causing the darker red areas.

The shift of these fits for all rows and spectral windows is then used to determine the collection 4 nominal wavelength map.
By determining the central column of each spectral window, a 2D polynomial is fitted through the shifted wavelength versus the row and column dimension. This fit is then evaluated for all columns, to yield a nominal wavelength map for all unbinned band pixels. After trying out different polynomial orders for both dimensions separately, the optimal combination proved to be a 2nd degree for the row dimension, and a 3rd degree for the column dimension. The residuals are mostly in the order of 5 pm, which is approximately the attainable accuracy of the fits with the solar reference spectrum due to limited spectral sampling
distance. Note that the wavelength key data in collection 4 is a pixel map with actual annotation data, and not a polynomial as in collection 3.

### 6.2.2  Wavelength temperature correction

During the operational mission lifetime of OMI, the optical bench temperature of the instrument has been steadily increasing from about 264 K up to 265.5 K, at which point the instrument thermal configuration was changed as described in Sect. 2. Due
to thermal deformations of the OPB, the wavelength associated with the detector pixels changes. To correct for this, a thermal wavelength calibration has been performed using binned irradiance measurements, as they are more prevalent throughout the mission. The average OPB temperature during the irradiance measurements of each orbit is plotted versus orbit number in Fig. 13, for all orbits included in this analysis during the operational part of the mission.

Using the same Levenberg–Marquardt fit method with the convolved high-resolution solar spectrum as applied above for the
nominal wavelength map calibration, the wavelength shift in the irradiance spectrum for each channel separately is determined using data from a selection of orbits. In Fig. 14 this wavelength shift is plotted versus OPB temperature for the UV1 channel as





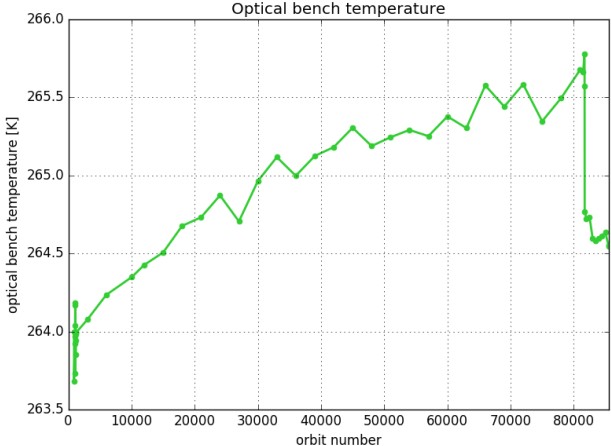

**Figure 13.** Average OPB temperature during irradiance measurements versus orbit number. The temperature can be seen to be steadily increasing during the mission lifetime, then decreasing at the end as a result of the new thermal configuration.

an example. Segmented linear fits are made through the operational temperature group, and from there to two higher commissioning temperature groups of points. Furthermore, a clear linear wavelength shift relation versus column number is observed as well in the fit results, therefore resulting in a stretch of the spectrum. The average slope of these linear fits is determined for the temperature groups as well and annotated in the CKD. The L01b processor linearly interpolates between these values for each irradiance measurement's OPB temperature and column number.

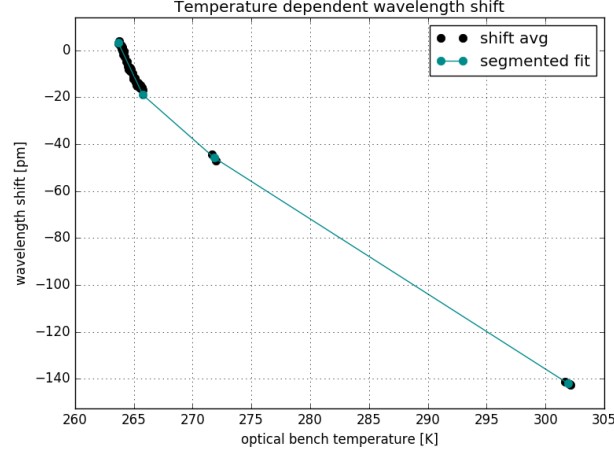

**Figure 14.** Observed wavelength shift in the UV1 channel for all orbits included in this analysis versus temperature as black dots. The dense group on the left is obtained during the nominal operation phase, whereas the two sparse groups on the right are measured early in the mission during the commissioning phase where the instrument was operated at higher temperatures.





### 6.2.3 Wavelength inhomogeneous slit illumination correction

In radiance mode, measurements are made with an integration time of approximately $2\,\mathrm{s}$ while the platform is moving with approximately $7\,\mathrm{km/s}$ in the flight direction. This combination causes rapid scene changes when flying over cloud edges, which results in sharply differing illumination of the spectrometers' entrance slit. This in turn leads to wavelength shifts during the measurement that are obscured by the internal co-adding of the instrument. The wavelength change due to this inhomogeneous slit illumination can be qualified with a so-called Q-factor that is derived from the small pixel column radiance data as described in the OMI ATBD (Ludewig et al., 2021). These small pixel columns are available without co-addition, and the Q-factor is a relation between the radiance of the first and last frame of a co-addition. A clear linear relation for the Q-factor between the UV2 and VIS channel has been found (UV1 has no small pixel column), which is applied when the small pixel column values of one of the bands have no valid data.

To determine the wavelength shift of the *radiance* measurements, the ozone absorption spectrum must also be taken into account. The Levenberg–Marquardt fitting method is extended to fit radiance measurements by also introducing this ozone spectrum with a fit parameter. The fit parameters utilized are now the wavelength shift, the solar intensity, the ozone intensity, background, slope and the derivative of the slope. This analysis is performed for all channels separately. The result of the fit is given in Fig. 15 for all UV2 pixel radiance values of 4 orbits spread out over 2005. The radiance wavelength shift is plotted versus the inhomogeneous slit illumination Q-factor, grouped per row. A linear fit is made through the data for each row and its slope shows a relation with row number; increasing at the edges of the band and decreasing towards the middle. Similar results are obtained for the VIS channel, not shown here. In contrast to the analysis for collection 3, the wavelength shift with respect to the Q-factor as determined in the collection 4 analysis stays constant with respect to the spectral fit windows and is therefore averaged over the spectral dimension for all detector columns.

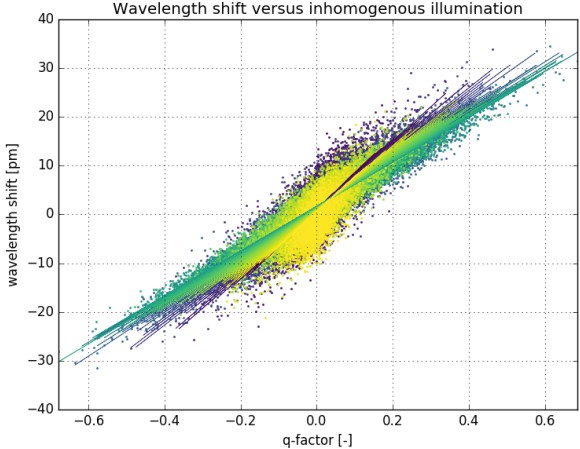

**Figure 15.** Scatter plot of the radiance wavelength shift for all UV2 pixel radiance values of 4 orbits spread out over 2005, versus the inhomogeneous slit illumination Q-factor. The colors indicate different rows.





This analysis provides the collection 4 CKD in the form of a slope and offset for each row, which in combination with the slit illumination Q-factor based on the radiance measurement small pixel column, determines the applied wavelength shift.

### 6.3 Geolocation line-of-sight

The geolocation azimuth and elevation line-of-sight (LOS) angles have been derived anew, but are based on the original 2004 collection 3 on-ground calibration. For each band the angles are averaged for each row over the columns in the detector science region. In collection 3 this row-averaging was done by the processor in the LOS binning algorithm. For the collection 4 processor it was decided that this step is done during the CKD creation process, because it is always the same operation and always yields the same results. The binning of multiple rows is still handled in the LOS binning algorithm depending on the 645 binning scheme.

### 6.4 Solar eclipse flagging

In collection 3, solar eclipses were flagged using a square latitude / longitude bounding box, and a time interval spanning the whole eclipse event during that specific day. This resulted in a large amount of ground pixels that were flagged unnecessarily. For collection 4 this was changed to a real-time geometrical calculation, thus only flagging ground pixels that are actually 650 within the shadow of the eclipse during each measurement duration. For a description of the theory and implementation of the algorithm in the processor, the reader is referred to the OMI ATBD (Ludewig et al., 2021). The calibration key data needed for this algorithm are the instant of greatest eclipse as Julian Date (JD), the Besselian elements and the time stamp $t_0$ relative to which these elements are defined. This data is taken from the NASA solar eclipse website for all eclipses of the years 2000–2100. Based on this data, the start and end times of each eclipse are determined beforehand, and included in the CKD 655 as well. These timestamps are relative to $t_0$ and are used as a first filter to see if a measurement falls within a solar eclipse period or not. If this is the case, the complete geometry is determined for each measurement time stamp and each ground pixel situated in any solar eclipse shadow type is flagged accordingly.

### 6.5 Transient signal flagging

The algorithms to detect transient events in the measured signals have been revised for collection 4. Here only the major 660 differences between collection 3 and 4 will be highlighted, for the full description of transient events and the theory and implementation of the algorithm in the processor, the reader is referred to the ATBD (Ludewig et al., 2021). The main differences between the collection 3 and collection 4 detection algorithms can be listed as follows:

- The collection 3 method only compared a measurement with the previous measurement in time; the collection 4 algorithm uses the current and both the previous and next measurement in time, something which is enabled by the new 665 processor architecture.

- The collection 3 method was based on a division with the previous measurement, while the new method subtracts the maximum of two adjacent measurements.



The concept for background and calibration measurements (like LED and WLS), where the signals are relatively stable in time, is similar for both the collection 3 and collection 4 algorithm and key data. However, the working principle behind the collection 4 algorithm for radiance and irradiance measurements, that are not stable over time, is completely different. This new method for radiance and irradiance measurements estimates the expected signal for a pixel by scaling the spectrum with respect to adjacent measurements in time, and then takes the median of the scale factors multiplied by the adjacent signal of the estimated pixel. This estimated value is then subtracted from the signal to form the jump, and the signal is divided by the estimated value to form the jump factor. These two results are then subjugated to their own thresholds, whereby pixels with estimated signals smaller than the jump threshold have their own lower jump threshold, with no jump factor threshold.

The signal jump for calibration measurements is very constant for non-transient pixels in time, and thus transient pixels can be filtered out easily. The thresholds here are set such that almost no false positives are flagged and virtually all transient pixels are flagged, which mean almost no false negatives. For radiance and irradiance measurements however, this is a bit more of a trade-off. Although the new method greatly improves the performance over the previous method, the thresholds have to be set in a safe manner so as not to flag too many non-transient pixels which show relatively large natural jumps in signal. The thresholds are set such that almost no false positives occur, while flagging most of the transient pixels.

## 6.6 South Atlantic anomaly

The South Atlantic anomaly (SAA) is a geographical region over South America and the South Atlantic ocean where cosmic particles can penetrate the protective shield formed by the Van Allen belts. In this region disturbances in the form of spikes and transient events are far more likely due to the presence of this relatively large number of cosmic particles that could hit the detector.

In collection 3, the SAA flagging region was based on a rectangular latitude / longitude bounding box. All measurements performed within this boundary were flagged as SAA warning, which flagged too many pixels because the SAA region is not square but almond-shaped. The collection 4 flagging algorithm has therefore been updated with a version that can handle arbitrary polygon regions. That is, an area can be defined as any set of polygon coordinates in longitude and latitude in the CKD as long as they form a closed loop. Using this polygon, fewer measurements are flagged unnecessarily.

The collection 4 region has been determined by analyzing 2 full repeat cycles: one in the end of 2005 (after in-flight calibration), and one in the beginning of 2019 (recent part of the mission). The L0 data for these $2 \times 233$ orbits was processed by the collection 4 L01b processor for background and radiance measurements until after the transient flagging algorithm. Radiance measurements were chosen as they are the only type of measurement performed constantly and globally. Band 1 (UV1) was chosen because this wavelength region is the most sensitive to transients due to the low UV radiance values at detector-level. Based on the number of transient pixels flagged in the radiance measurements of band 1 combined with the satellite position, a transient density map was determined.

In Fig. 16 the collection 3 rectangular bounding box limits are shown based on an analysis of data from 2004, which was later improved in 2007. These square SAA limits are plotted together with the polygon contours of the analysis discussed above. The center-points of the contours based on the repeat cycles of 2005 and 2019 are determined and indicated by a dot of



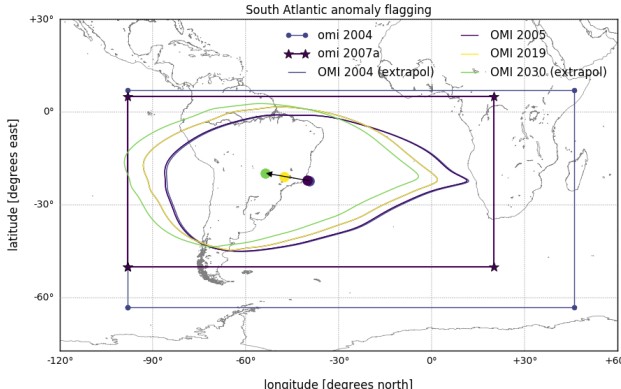

**Figure 16.** SAA bounding box limits for collection 3 are shown as the square regions based on the 2004 (light-blue) and 2007 (purple) analysis. For collection 4 polygons were derived using 2005 (blue) and 2019 (red) data. The centerpoint of the polygon contour is indicated by a dot of the same color, and clearly drifts in time. The movement of the centerpoints is used to extrapolate the contours down to 2004 (light-blue) and up to 2030 (yellow).

the same color. It turns out that the SAA region is slowly drifting westward in time, and it is favorable to take this effect into account in the flagging algorithm. The movement of the center-point is used to extrapolate these SAA regions to orbit 1 (start of mission) and 150000 (well past end of mission). Within the repeat cycles, all contour points are interpolated with their own

average movement between the repeat cycles.

### 6.7 Row anomaly flagging

The row anomaly is thoroughly described in Schenkeveld et al. (2017); here only the analysis needed to obtain calibration key data for the collection 4 L01b processor will be discussed.

To monitor the row anomaly, collection 4 L1b radiance data was analyzed for indicators that show which rows have dis-

710 turbed measurements. It was found that the radiance averaged over all columns shows very stable signal values in the scanline dimension for anomaly rows, while normal measurements are more fluctuating. Therefore, a monitor was set up that determines for each scanline the running standard deviation for 300 neighboring scanlines. The measurements for all radiance modes are merged and fill values are filtered. Furthermore, ground pixels with a solar zenith angle larger than 90° are filtered, because the signal is not strong enough yet at the very beginning and end of the orbit. The UV1 band shows a large number of saturated

pixels at the end of the orbit due to the row anomaly that disturb this indicator, so these are filtered out as well. The result is shown in Fig. 17 for the VIS channel where a certain row range clearly shows very low values.

Another indicator was found in the radiance monitor fit wavelength shift parameter, which is averaged over all fit windows after which the absolute value is taken, while applying the same filters as above. Because the result is still quite noisy, the moving average for each scanline is calculated based on 100 neighboring scanlines. The result can be seen for the VIS channel





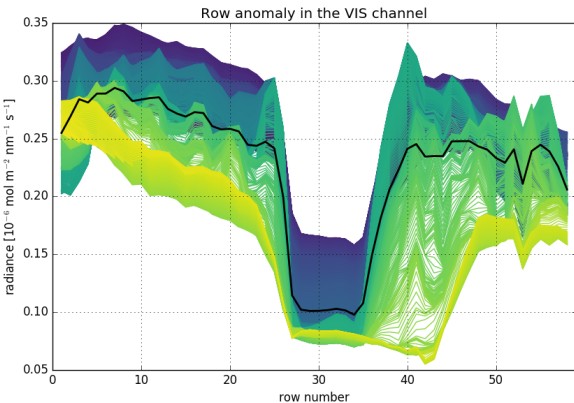

**Figure 17.** Row anomaly monitor results for the VIS channel. Here the radiance data from orbit 50000 is averaged over all columns, after which the running standard deviation is determined over 300 neighboring measurements in time. The colors indicate the sequence of measurements in the orbit, starting with dark blue over the South Pole, and changing via green to light yellow towards the end of the orbit over the North Pole. The figure shows low values for anomalous rows for the whole duration of the orbit, while the disturbed row range increases towards the end of the orbit.

in Fig. 18 where it is clear that some rows show extreme wavelength shifts due to the row anomaly, for most of the orbit. The effect is visible for the other bands (not shown here) too, although less clear.

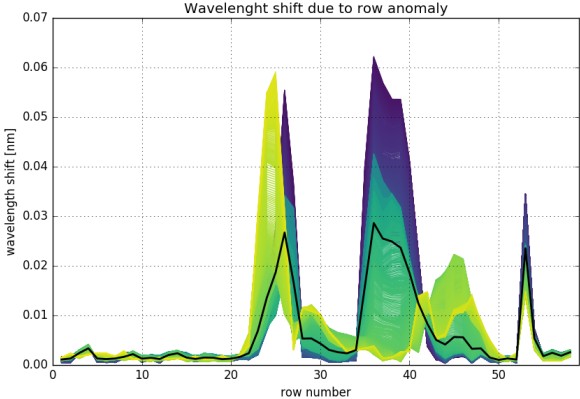

**Figure 18.** Radiance monitor fit wavelength shift in the VIS channel obtained from orbit 50000. The data is averaged over all fit windows and the absolute value is taken. Then for each measurement the moving average is determined for 100 neighboring measurements. The color scheme is the same as in Fig. 17. The result shows very high shift values for anomaly rows for most of the orbit.

Row anomaly data for the whole mission has been analyzed for the UV2 and VIS channels. These have been determined for each day and two wavelengths per channel. Based on these results a dynamic map is created which the collection 4 L01b





processor uses to flag rows accordingly in time. This dynamic map is updated whenever needed to reflect the actual status of
the row anomaly.

## 6.8  Digital elevation map

In collection 3 OMI L1b products, terrain height and a surface classification flags are written based on the NASA 90 arcsec
digital elevation model (DEM). This DEM is based on the global multi-resolution terrain elevation data 2010 (GMTED2010)
model (Danielson and Gesch, 2011).

These collection 3 values are determined at the centerpoint of the OMI ground pixels. Because the resolution of the 90
arcsec DEM is much higher than the resolution of the OMI ground pixels, the method to use the values at the centerpoint is not
very accurate, especially not in mountainous areas. Therefore, for collection 4 products a method has been used that has also
been used for TROPOMI L2 products. Instead of using the value at the centerpoint, the averaged value of an area around the
centerpoint of the ground pixel has been used. To do this efficiently a new DEM was created with these averaged terrain heights
for different area sizes, still using the 90 arcsec data. For the shape of these areas circles are used with diameters of 10, 20, 30,
40 and 50 km. Besides the averaged terrain height, also the standard deviation and minimum and maximum values are stored
in this new DEM. Furthermore, the dominant surface classification (land), and several types of water (inland water, shallow
ocean, deep ocean, etc.) and the water fraction are stored. The L01b processor utilizes this new DEM and for each ground pixel
the DEM area size is selected that comes closest to the actual area size of the ground pixel, and the corresponding values of the
DEM point that comes closest to the ground pixel centerpoint are written to the L1b product. In Fig. 19 the improvement over
mountainous areas is clearly demonstrated.

## 7  Unchanged calibration and algorithms

For completeness all algorithms and calibrations that are unchanged between collection 3 and 4 are summarized here:

- The limits for pixel full well, ADC overflow and co-addition overflow have not been changed. Note that the limits for
register full well did change as described in Sect. 4.3.

- The parameters and limits for the electronic saturation flagging algorithm have not been changed.

- The read-out noise annotation of the detector has not been changed.

- The background correction key data for collection 3 used to be determined outside the L01b processor, in the TMCF
      system. With the update to collection 4, the correction is determined by the L01b processor, using a similar algorithm.
The background analysis algorithm in the L01b processor is combined with the RTS analysis algorithm.

- The dark current temperature coefficients have not been changed.

- The row and frame transfer times have not been changed. This means that also the smear correction remains unchanged,
      apart from a different handling of flagged pixels.





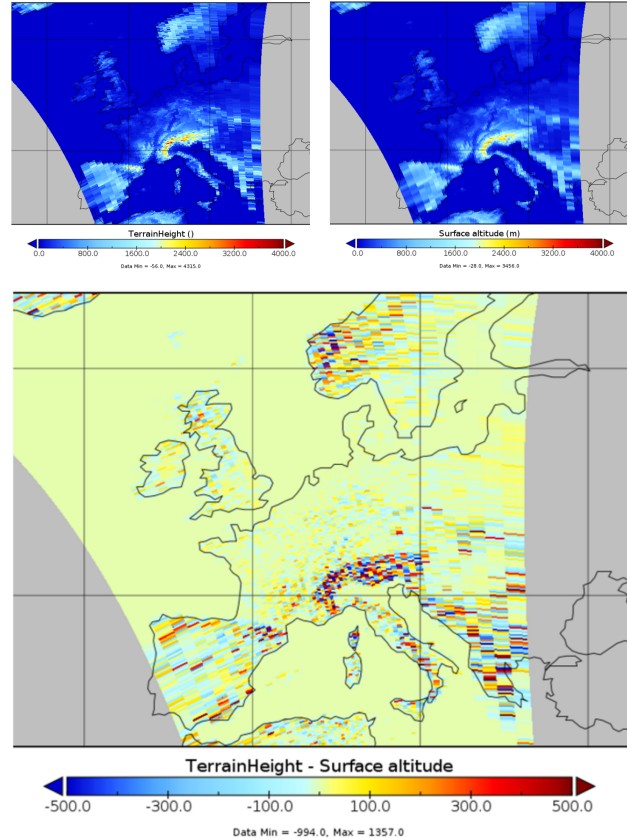

**Figure 19.** The top left figure shows the terrain height for orbit 84291 in the collection 3 product. The top right figure shows the surface altitude for the same orbit in the collection 4 product. The bottom figure shows the difference, especially in mountainous areas, such as Norway and the Alps, the differences are clearly visible.

    – No potential improvement has been found for the pixel response non-uniformity (PRNU) and slit irregularity calibration.

– The straylight correction is applied in the same way as in collection 3, apart from a different handling of flagged pixels.

For all details on the updated and unchanged algorithms the reader is referred to the OMI collection 4 L01b ATBD (Ludewig et al., 2021), also provided as supplemental information to this paper.

## 8 Radiometric verification

As the choice was made not to change the day-one radiometric calibration of the instrument, it is assured that the validation
of the radiometry has not changed with respect to the results presented in Dobber et al. (2006) and Dobber et al. (2008a). The





following sections are to present verification that all intended changes between collection 3 and collection 4 are implemented, no unintended changes have occurred, and that all differences between the two collections are understood.

An apparent enhancement in sensor response between 350 nm and 380 nm compared to surrounding wavelengths (see Fig. 4 and Fig. 9) is very likely caused by a change in the dichroic filter used to separate the UV2 and VIS channels. A shortward
shift of the filter response curve explains the observed solar and Antarctic signal changes quite accurately. In collection 4 this approximately 1% effect is ignored. A similar 1% feature has also been observed between 300 nm and 310 nm in the UV1 channel, though only in the measured irradiance changes. The cause of this anomalous change is unknown and it too has been ignored in the collection 4 calibration.

The verification results for the L1b irradiance, the instrument BSDF, and radiance are described in Sect. 8.1, Sect. 8.2 and
Sect. 8.3, respectively.

## 8.1 Irradiance verification

Expected differences between collection 3 and collection 4 irradiance data are the fact that in collection 3 the irradiance is not corrected for degradation and that the flagging of bad pixels is more aggressive. In addition, a bias is expected due to the Earth-Sun distance normalization that is present in collection 4 and not in collection 3. This normalization has been undone for
easy comparison here.

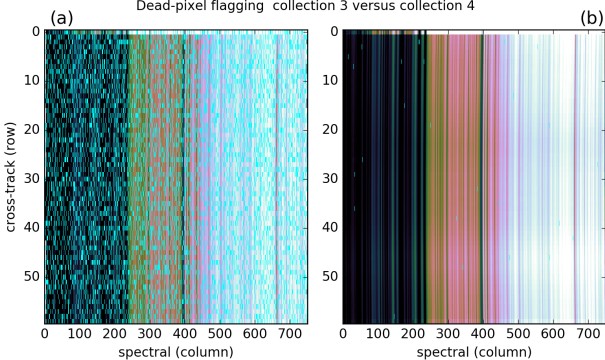

**Figure 20.** The VIS irradiance spectrum for all detector rows, shown as image for orbit 89169 for collection 3 (left) and collection 4 (right). The cyan dots in the left panel are fill values in the L1b data, caused by the too aggressive dead-pixel flagging.

As can be seen in Fig. 20 there are a lot of pixels with fill values in the collection 3 irradiance spectrum due to aggressive flagging of bad pixels in collection 3. The comparison of UV2 and UV1 channels shows similar differences between collection 3 and collection 4 irradiance data.

In Fig. 21 the collection 4 irradiance data is plotted versus the collection 3 irradiance data. The missing pixels due to the
aggressive flagging in collection clearly show up as gaps in the figures, that disappear in the collection 4 spectra. The Earth-Sun-





distance effect is removed from the figures, and the difference ratio shows the effect of the irradiance degradation correction, which has a smooth spectral dependence, stronger to the shorter wavelengths, and in-line with the results as given in Fig. 7.

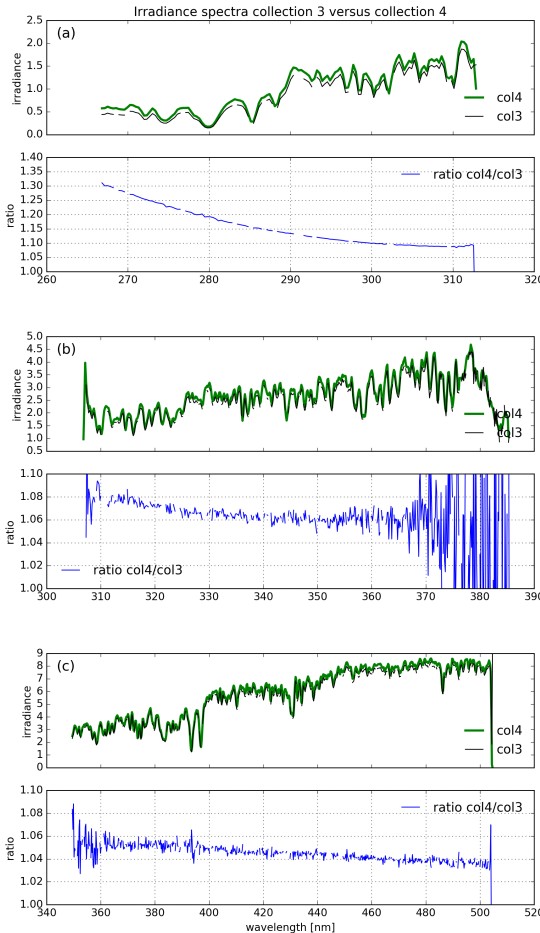

**Figure 21.** Comparison of the irradiance spectra between collection 4 and collection 3 for the UV1 (top), UV2 (middle) and VIS (bottom) channels. In each plot the top panel shows both absolute values, and the lower panel shows the ratio between the two collections, as observed late in the mission for orbit 88854.

## 8.2 BSDF verification

The reflectance is the ratio between the incoming sunlight and the reflected light from Earth. The BSDF is the relation between

how the instrument perceives the solar radiation (irradiance) and the radiation coming from Earth (radiance), and is defined as the ratio between ABSRAD and ABSIRR, being the absolute radiance and irradiance conversion factors, which define the absolute radiometric calibration of the instrument. Due to the degradation of the instrument the response of the instrument





changes, as a function of time (or orbit number) and the detector pixel location. This degradation is compensated for using the degradation corrections for the Earth and Sun ports, as it would otherwise be introduced into the observed reflectance.

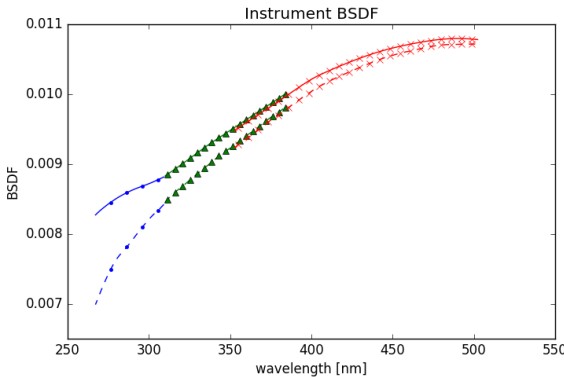

**Figure 22.** Instrument BSDF using the QVD diffuser for the UV1 (blue dots), UV2 (green triangles) and VIS (red crosses) channels respectively. It can be seen that the BSDF matches quite well in the overlap regions between the bands at the start of the mission (solid line) and late in the mission (dashed line).

In Fig. 22 the BSDF over the three bands are plotted as a function of the wavelength for the central row. The initial day-one BSDF is shown together with the apparent BSDF later in the mission, based on the separate degradation corrections for Solar irradiance and Earth radiance. This shows that the BSDF is, first of all, a smooth function of the spectral parameter, and second that in the overlap regions it was matched, and remains matched when degradation is taken into account.

     In Fig. 23 the row dependence of the BSDF at the start of the mission and after 17 years in orbit is shown. Clearly the BSDF
of the instrument has changed over time, and this effect depends on the row dimension too. Due to these corrections in the L01b processor, the errors, otherwise introduced into the Earth's reflectance, are mitigated as far as possible.

### 8.3   Radiance verification

It is expected that the radiometric differences in radiances between collection 3 and collection 4 do not depend on wavelength, but only on row number because no spectral corrections were made in the degradation correction. This must be the case
because we have forced the BSDF wavelength dependence to agree with the irradiance wavelength dependence. Note that in this comparison we have corrected for the different handling of the Earth-Sun distance in the two collections. In Fig. 24 the differences in radiance is shown for all three channels, in absolute terms and as a ratio. As can be seen there is a 2.5% bias due to the radiance degradation correction, and no spectral dependency, as intended. The value is in line with what to expect for this orbit number (84293) based on the results given in Fig. 9. In the figure some small spectral structures seem to appear, but
these are caused by the different approach to wavelength assignment. Due to these small differences in the wavelength scale, interference patterns as shown in the figure will occur around spectral lines.



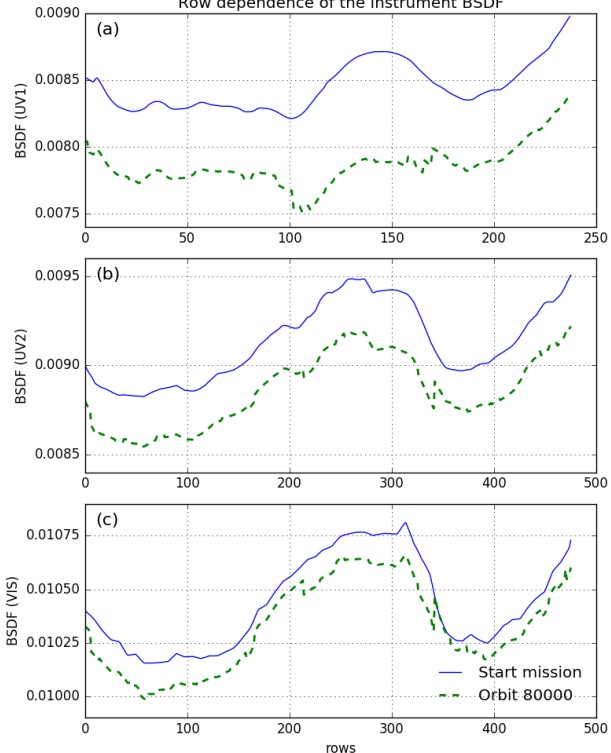

**Figure 23.** Instrument BSDF using the QVD, showing the row dependence of the BSDF for UV1 (band 1, panel (a)), UV2 (band 2, panel (b)) and VIS (band 3, panel (c)). The blue solid lines shows the BSDF at the start of the mission, in green dashed line after 17 years in-flight.

Note that for the VIS channel (bottom plot) a jump appears for the signal below 360 nm. At this specific column a gain ratio switch occurs in the specific radiance observation mode. In collection 3 no corrections were made for drifts in the gain ratios, but in collection 4 these are taken into account as described in Sect. 4.2

## 9 Conclusions

A new collection 4 dataset for the OMI mission has been established to supersede the current collection 3 Level 1b data series. This dataset is produced with a newly developed L01b data processor based on the TROPOMI L01b processor. The collection 4 L01b processor is running in the forward stream at the NASA OMI SIPS since April 2020, and the reprocessing of the entire 17 year mission up until now is in progress. The collection 4 L1b data has a similar output format as the TROPOMI L1b data, for easy connection of the two data series. Many insights of the TROPOMI algorithms were included, as well as insights learned from the usage of OMI collection 3 data.

A significant improvement over collection 3 is the detrending of instrument effects and optical degradations. Drifts in electronic gain are now corrected for, and pixel quality flagging has improved strongly. The TMCF system is not required anymore

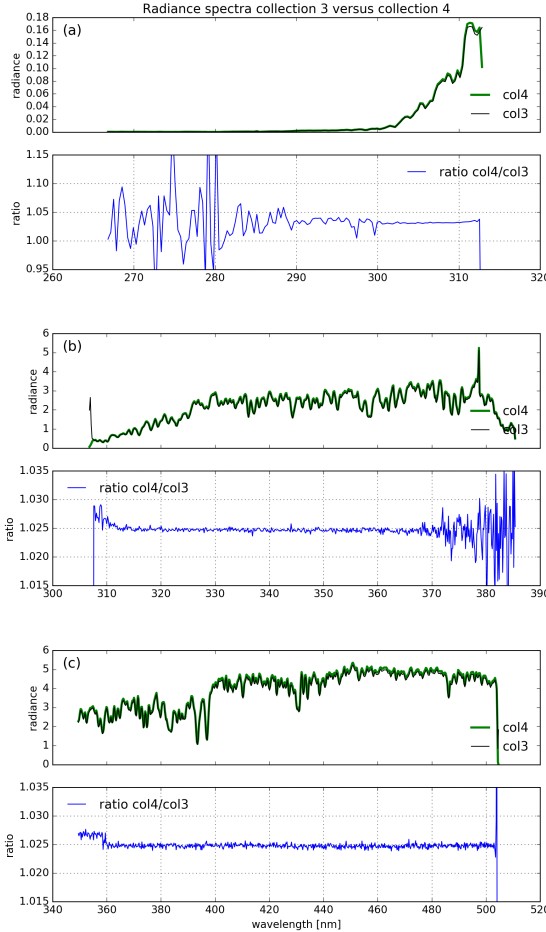

**Figure 24.** Comparison of the radiance spectra between collection 4 and collection 3 for the UV1 (panel (a)), UV2 (b) and VIS (c) channels. In each plot the top panel shows both absolute values, and the lower panel shows the ratio between the two collections, as observed late in the mission for orbit 84293.

for background correction because this is now included in the L01b forward processing, as is RTS flagging. The initial radio-
metric calibration from day-one has been re-used, while the optical degradation of the instrument BSDF has been corrected for, such that the observed Earth reflectance is not affected by instrumental artifacts. While the radiometric drift corrections are certainly an improvement over collection 3 data, which had no such corrections, they are not equally robust at all wavelengths. In deriving the instrument irradiance calibration the solar output was assumed constant, and the radiance calibration technique is only valid above 330nm. In both cases the estimated instrument change at shorter wavelengths, especially those in the UV1
channel, carries enhanced uncertainty. It is unlikely, for instance, that UV1 data could ever be used to measure accurate ozone trends. Many improvements have been included in the annotation data and the flagging data.



It has been verified that all changes are as intended, and that the resulting L1b data is a clear improvement of the previous collection 3 dataset. In parallel, updated collection 4 L2 data processors are under development. These are based on the most recent TROPOMI L2 processors, such that the 17 year OMI data record can consistently be connected to the data series from its successor TROPOMI.


*Data availability.* The collection 4 data products (OML1BIRR; OML1BRUG; OML1BRUZ; OML1BRVG; OML1BRVZ) described in this paper are publicly available through NASA GES DISC.

*Author contributions.* Nico Rozemeijer is the technical lead and architect for the OMI and TROPOMI L01b software development, and acts as the deputy project lead. Quintus Kleipool is the KNMI lead instrument scientist for OMI and TROPOMI, and acts as the project lead
for the development described in this paper. Mark ter Linden is the technical lead and architect for the OMI and TROPOMI L2 software development for the KNMI science products. Mirna van Hoek is lead instrument operations for the OMI and TROPOMI payloads, and performed the analysis for calibration of the irradiance angular dependency and the irradiance degradation and is also responsible for the software unit testing. Jonatan Leloux is the geolocation engineer and responsible for the annotation algorithms and calibration analysis. Erwin Loots is responsible for the L01b algorithm design and analysis of the calibration key data. Antje Ludewig is optical expert and provides
calibration analysis for the required key datasets and acts as the software test manager. Emiel van der Plas provides calibration analysis for the required key datasets and also responsible for the software system testing. Daley Adrichem is algorithm engineer for the L01b development. Raoul Harel is algorithm engineer and also responsible for the L01b software unit testing framework. Simon Spronk is algorithm engineer and responsible for the averaged L1b irradiance processor development. Glen Jaross is the NASA instrument scientist for L1b development. David Haffner is the lead of the NASA L1b team and supplied the radiance degradation correction data. Pieternel Levelt is the OMI Principal
Investigator (PI), while Pepijn Veefkind is the deputy PI.

*Competing interests.* The authors declare that no competing interests are present.

*Acknowledgements.* OMI was built by Dutch Space (now Airbus Defence and Space Netherlands) and TNO-TPD (now TNO Science & Industry) in The Netherlands, in cooperation with Finnish subcontractors VTT and Patria Finavitec. The instrument was financed by the Netherlands Agency for Aerospace Programmes NIVR (now Netherlands Space Office NSO) and the Finnish Meteorological Institute (FMI).
The work described in this paper was funded by the NSO. The US investigators acknowledge support from NASA's Earth Science Division through the Aura project, operations and science team (NNH19ZDA001N-AURAST) managed by Ken Jucks and Barry Lefer. The authors wish to thank Phil Durbin for his support throughout the OMI collection 4 development including the testing and integration of the software at NASA OMI SIPS. The many contributions of OMI science product developers in testing and providing feedback on the Collection 4 data sets is also gratefully acknowledged. We thank Matthew Bandel, Sergey Marchenko, and Zachary Fasnacht for additional testing and
verification of the results in this paper. The authors also acknowledge the support of James Johnson and Irina Gerasimov for the distribution



of the data product through the NASA GES DISC. The authors wish to thank the OMI science advisory board members Pieternel Levelt, Joanna Joiner, Johanna Tamminen and Pawan Bhartia for their continued support throughout the development of OMI Collection 4.



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
