# Peer review of "OMI Collection 4: establishing a 17-year long series of detrended L1b data"

_Atmospheric Measurement Techniques, 2021_

## Author Response (AR1)

**Review comment amt-2021-430-RC1**

**Reviewer: Anonymous Referee #1**

Dear referee,
Thank you for your detailed review of our article. Our responses to your remarks, questions and considerations can be found in the tables below. The responses also include the planned actions for the revised manuscript.

**Response to specific comments**

| Item | Referee comment | Author's response |
|---|---|---|
| Page 1/ line 6 | '… until the eventual end of the mission.' Is this to be understood including possible extension even beyond 2023? In line 3 you mention: 'for many years more.' Suggestion to be more specific and possibly state the limitation on the extension of a mission, which is most likely here also the case due to remaining necessary fuel for deorbiting. Later you mention this on page 4/ line 91,92. | Agreed, we will be more specific |
| Line 8,9 | In combination with the title, is the assumption correct, that data of the past 17 years is planned to be reprocessed? Maybe worth mentioning in the abstract already, if reprocessing is planned/ done. Later in the conclusions page 38/line 814 it is mentioned: 'the reprocessing of the entire 17 year mission up until now is in progress'. | Agreed |
| Line 17, 18 | Is the understanding in combination with the statement in line 3 correct, that TES and HIRDLS are not operated anymore? Suggestion to state more explicitly the current status of TES and HIRDLS. | Agreed, we will add more information. TES was decommissioned 32/01/2018 and HIRDLS stopped working 17 March 2008. |
| Line 17, 18 | '..instrumental effects that are common' suggestion to state the main differences between the optical paths between sun and Earth port, e.g. diffuser. | Agreed |
| Line 48, 49 | 'For collection 4 the TROPOMI naming convention was adopted, referring to the UV1, UV2 and VIS channels as band 1, band 2 and band 3 respectively.' Can you add an explanation why this has been adopted? | Agreed, will change the text to clarify why the terminology was chosen. |

| Item | Referee comment | Author's response |
|---|---|---|
| Page 3/ line 54, 59 | Suggestion to add references for collection 1 and collection 2 dataset, e.g. Oord, 2006 SPIE and Oord, 2006, IEEE, vol 44, no 5, see also page 6/ line 154 where one of the references is provided, but here for collection 3, which was earlier referenced to Dobber, 2008. | Agreed, we can give more details/references on earlier collections |
| Page 4/ line 114, 115 | you mention completely understandably, that the updates of the KNMI and NASA L2 processors fall outside the scope of this paper, but could you possibly add some references? | There are no publications yet for the updated L2 OMI processors. |
| Page 5/ line 132 | To get a better understanding what 70 000 orbits mean in time, could you add in the introduction to OMI, how many orbits per day OMI performs, e .g. around page 2, paragraph starting at line 30? | Agreed |
| Page 9/ line 240 / section 3.5 | Might it be, that an angular dependence correction is nonoptimal leading to this 'striping'? Is a seasonal effect observed? Suggestion to also add a figure to illustrate this observed effect. | We will add a suitable reference to the striping. The subject is rather intricate and we would not be able to do it justice in a sentence or a single plot. |
| Page 15, 16/ section 5.2 | It may be worthwhile stating, that even if the QVD degraded more than the ALU diffusers, the degradation shown over those 12 years (table 4), 16 years page 17 (figure 4) is very low compared to other instruments, especially considering its daily use. | Agreed |
| | Generally not for all described changes to the processor from collection 3 to collection 4 the improvements are described/ shown by absolute, error bar reductions or end-product improvement. Here some examples:  Suggestion to amend graphs and/or values of significant improvements, where missing. | Not all improvements can be directly compared  with collection 3,  we will add more information wherever possible. |
| Section 6.2 | 'Furthermore, this over-fitting can result in unexpected behavior for extreme values of other input variables like the OPB temperature as well.' but no numbers provided and improvement not clear; | We will rephrase this to make the improvements clearer. |
| Section 6.1 | what is the advantage of the difference implemented in collection 4? | The wavelength calibration is now reduced to the annotation which is used by L2. Will clarify this point in the text. |
| Section 6.2.1 | improvement of changing the method on end-product not clear; | We will clarify the benefits of the pixel map in the text. |

| Item | Referee comment | Author's response |
|---|---|---|
| Section 6.2.3 | Improvement not clear | We will clarify the improvements. |
| Section 6.3 | other method described, but improvement not clear; | The calculation is not performed with every processor run but implemented as calibration key data. This makes the processor more efficient. We will clarify this point. |
| Section 6.4 | 'resulted in a large amount of ground pixels that were flagged unnecessarily' without giving e.g. percentage improvement; | We will give an estimate on the improvement. Note that the improvement depends on the shape of the specific solar eclipse and up to 90% of the pixels were flagged unnecessarily. |
| Section 6.5 | transient signal flagging | We will add examples on the occurrence of transient flags. |
| Page 35/ line 774 | 'bias is expected due to the Earth-Sun distance normalization that is present in collection 4 and not in collection 3.' If understood correctly a bias is introduced by the different method in collection 4. And, the bias is basically the improvement implemented by the new correction, but not shown in comparison with the former data from collection 3. Previously on page 8/ line 213 it is only stated that now both radiance and irradiance are corrected for Earth-Sun distance. Please consider to make the text more explicit. And please describe the value of the bias which is understood as the improvement in collection 4. | This part is not phrased very clearly, we will improve this. The collection 4 now includes a correction for Earth-Sun distance. When comparing to collection 3 this step needs to be removed to allow for an unbiased comparison. |
| Line 776, 777 | Is the mentioned 'aggressive flagging' linked to page 12/ line 310 section 4.5 Detector pixel quality flags? If yes, suggestion to add reference to that section here. | Thank you, we will add the reference to the section. |
| Page 39/ line 821 | 'that the observed Earth reflectance is not affected by instrumental artifacts' might this be a too strong argument, since also the text describes there remain some effects, which are not able to be identified in flight, e.g. folding mirror, telescope mirror? Suggestion to change the wording slightly, e.g. is 'not significantly affected'. | We will change the wording. |

**Response to technical corrections**

| Item | Referee comment | Author's response |
|---|---|---|
| Page 3/ line 67 | trend and calibration monitoring system (TMCF)' is it TCMF or trend monitoring and calibration system? | The latter is correct, we will change the text accordingly. |
| Page 9/ line 237 | CKD file, please provide abbreviated text. | Agreed |
| Page 15/ line 394, 395 | QVD, quasi volume diffuser ALU1 and ALU2 diffusers made from aluminium. | Agreed |
| Page 16/ line 403 | 'ratio From' à ratio. From | Agreed |
| Page 36/ figure 21 | suggestion for visualization to use the same y-scale for the ratios from 1.00 to 1.40 as for the UV1 for all channels and to use dots instead of lines for better visibility and comparison. | We will improve the plot. |
| Figure position | The figures positioning sometimes interrupts a sentence of the text, or , e.g. page 34/ figure 19 are placed in the next section. Consider repositioning the figures closest to their description in the text. | This is partly an effect of the latex template and the used manuscript style. We will try to improve the positioning. |
| Last but not least | Maybe it would be nice to refer also to one of the early OMI papers by its optical designer Huib Visser, e.g. Smorenburg, C., H. Visser, and K. Moddemeijer, "OMI-EOS: Wide field imaging spectrometer for ozone monitoring", Europto/SPIE conference, Berlin, 1999, SPIE volume 3737, 1999 and/or Piet Stammes, Pieternel F. Levelt, Johan de Vries, Huib Visser, Bob Kruizinga, Kees Smorenburg, Gilbert W. Leppelmeier, and Ernest Hilsenrath "Scientific requirements and optical design of the ozone monitoring instrument on EOS-CHEM", Proc. SPIE 3750, Earth Observing Systems IV, (24 September 1999); https://doi.org/10.1117/12.363517. | Agreed, will add the latter. |

**Review comment amt-2021-430-RC2**

**Reviewer: Anonymous Referee #2**

Dear referee,
Thank you for your detailed review of our article. Our responses to your remarks, questions and considerations can be found in the tables below. The responses also include the planned actions for the revised manuscript.

**Response**

| Item | Referee comment | Author's response |
|---|---|---|
| Page 4 | "This updated OMI processor has in-orbit calibration functionality in forward mode, making the TMCF system obsolete. The available TMCF calibration data has been analyzed, such that historic trends in the instrument calibration status can be corrected for in the collection 4 L01b (re-)processing." | See below |
| Page 6 | "The instrument operation schedule has been updated such that calculation and calibration needed for background correction and random telegraph signal detection can now been done by the collection 4 L01b processor in forward mode without the need for the TMCF system."
"The design of the collection 4 L01b makes it possible to have dependencies between measurements and perform aggregate calculations."
"This allows, for example, to initially process background measurements, and use an aggregate of these processed background measurements in the background correction during the processing of the remaining measurements."
We assume that this processing approach is applied to one orbit only, but this is not clear from the text. Is it possible to also apply this approach to multiple orbits, or to measurements / results from several days / weeks / months? | We will clarify this point,  For the background correction the previous 24h of background data is aggregated. |
| Page 6 | "Another improvement is that the tables allow a more fine-grained processing configuration." It is unclear from the text if this refers to measurement class (as indicated), or to ICID (Instrument Configuration IDentifier). | We will clarify this point. |

| Item | Referee comment | Author's response |
|------|-----------------|-------------------|
| Page 9 | "For collection 4 L2 processing an alternative irradiance product is generated that consists of the running average over 100 daily irradiance measurements, yielding an improvement of the signal-to-noise ratio with a factor of 10." This requires a memory capability in the processing system. How is this implemented? | The irradiance averager is a separate post-processor. We will make this clearer in the text. |
| Section 4.6 RTS | In collection 3 the RTS map is based on analysing 30 days of dark signal data. In collection 4 one day of data is used. It looks like collection 3 is more looking more RTS in general, whereas collection 4 is more looking for RTS that is considered relevant for the L1b accuracy. It would be interesting to know and understand more about the differences between these 2 methods. | That is correct, with a background correction based on daily measurements, changes in RTS on a long time scale are already accounted for. Therefore only RTS behaviour which is faster than the updates for the background correction are flagged. We will add more explanation to the text. |
| Section 5.1 | "A small change however is that in collection 3 the sensitivity calibration, as used by the L01b data processor, was provided as a function of wavelength in the calibration key data. For collection 4 the TROPOMI convention was used, and the calibration key data was converted to be a function of detector pixel." How do you deal with wavelength shifts for collection 4? | Will add a cross reference to the wavelength annotation in Section 6.2, there also corrections for shifts are explained. |
| Figure 4 | - The caption refers to top and bottom panels instead of left and right panels. - "Clearly there is an overall 4% degradation with no strong wavelength dependence [ALU1]" This is surprising and seems to point to a non-optical origin, such as perhaps geometric or electronic effects. Please elaborate a bit more on the origin of this observed 4% wavelength-independent degradation. | We will correct the caption. We will elaborate more on possible causes (see also below). |
| Section 5.3 | Relative irradiance : It would be interesting to know more about the final accuracy differences between collections 3 and 4. | The relative irradiance is a multi-dimensional problem, so it is not straight forward to compare. We will give an indication of the changes. |
| Section 5.4, Figure 7 | The caption refers to upper and lower panels instead of left and right panels. | We will correct the caption. |
| Section 5.4 | "This suggests that 2% – 3% of the observed change is independent of wavelength and not a result of optical degradation. Also it is evident that the degradation can be strongly row dependent, especially for the UV1 channel." | We will discuss the possibilities for different types of instrument change to explain the observations. However, we lack the necessary information to pin down the exact cause of the wavelength-independent changes. |

| Item | Referee comment | Author's response |
|---|---|---|
|  | What is the expected cause of this 2-3% offset? Does it make sense to include this in the irradiance degradation correction, when the cause is not optical? What is the expected cause of this row dependency? |  |
| Figure 14 | The indicated wavelength shift is 140 pm over 40K. Please indicate how much this is in spectral pixel size (e.g. 0.13 spectral px). | Agreed |
| Figure 15 | The indicated wavelength shift is 60 pm over a Q-factor range of 1.2. Please indicate how much this is in spectral pixel size (e.g. 0.06 spectral px). | Agreed |

**Review comment amt-2021-430-RC3**

**Reviewer: Ruediger Lang**

Dear Ruediger,
Thank you for your detailed review of our article. Our responses to your remarks, questions and considerations can be found below. The responses also include the planned actions for the revised manuscript.

**General comment**

*"For the correction of the Earthshine part the authors apply a "stable ground target" approach, also used by other missions for this purpose, where the target surface reflectance can be expected to be stable over the year and atmospheric variability is not too large. The choice of the target by the authors is snow/ice surfaces over Antarctica. While those surfaces should be quite stable (although snow BRDF function can be changing in a complex way as a function of temperature and solar illumination conditions) I am wondering if this is actually a good choice for a mission where ozone is contribution to a significant extend to the spectral variation of the measurements, in particular below 350 nm. Variability of Ozone is very large over the year in Antarctica, and arguably much more significant than at mid-latitudes. While in the latter case line absorber variability is larger (and stronger) like water vapour, these are usually covering only a small subsection of the spectrum and can therefor much better be filtered out. So I would have considered the Libyan desert being a better target, with an even more stable surface over the year (and well characterised), and less interference by ozone variability. I particular, and as a consequence of the strong interference and variability of ozone below 335 nm, a correction of the radiances in this important region (with many level-2 products derived from this part of the spectrum) based on actual measurements, has not been carried out. Instead it has been assumed that the degradation is spectrally neutral for the Earthshine port, so can be based on the degradation coefficients derived in the region between 335 to 360 nm for band 2 and 390 to 500 nm in band 3. However, the exact regions considered usable and used for Earthshine degradation evaluation for target area measurements (and extended across the full spectrum I guess) are not explicitly stated, since other spectral regions are suffering from atmospheric absorption features, Fraunhofer lines and interference of a dichroic. I consider the assumption on spectral neutrality a critical one and I find it has not been addressed in full by the authors. The results presented for Earthshine port correction could potentially be significantly biased because of this assumption. On the other hand, the results derived from the AU1 and AU2 diffusers, which indicate that the spectrometer and the detector assembly's contribution to the degradation seems to be indeed spectrally quite neutral (and there can be physical arguments also made for such an observation) have not been explicitly applied to support the hypothesis, e.g. by comparing it to the observed degradation in the 335 to 360 nm region and make some interference from such comparison. But most important I think the first mirror, which seems to be bypassed by the solar port optical path, cannot simply be ignored, in particular in the case that the region below 335nm is not addressed directly by Earthshine observations. Obviously any mirror in the light path could exhibit relative spectral neutrality in its degradation in the visible while exhibiting a strong spectral dependence in its degradation for shorter wavelength. Acknowledging the fundamental difficulty in assessing the Earthshine port degradation in this shortwave spectral region, while at the same time also acknowledging the larger number of users of collection 4 data*

*using particular this spectral region, I would strongly recommend to include some (at least initial) analysis applying level-2 retrievals, or applying (ozone) cross-section spectral dependency information to support the assumption on spectral neutrality."*

**Response to general comment**

We thank the reviewer for his thoughtful commentary on our approach to long-term radiometric calibration of OMI, and rather agree with his conclusion that the uncertainty in our assumptions has not been fully addressed. The reviewer is correct that all scene stabilization techniques in the UV are limited due to ozone absorption. This problem is universal, though it is more pronounced in high latitude ice surfaces due to the increased atmospheric path length. The high degree of predictability of such ice surfaces counterbalances the increased uncertainty due to ozone. Jaross et al. 2008 establishes a 2% absolute uncertainty in the technique for wavelengths as short as ~330 nm. Because we include ozone absorption in our atmospheric model this uncertainty is mostly a result of uncertainty in the surface BRDF and not caused by ozone variability. All ozone instruments for which we have used this technique derive ozone concentrations using wavelengths shortward of 330 nm with much larger ozone sensitivity. The ozone-related error at 330 nm is therefore a second-order effect.

Furthermore, we are not attempting to establish or verify an absolute radiometric calibration, rather a time-dependent calibration. The viewing conditions over Antarctica for a stable orbit are very repeatable each solstice. The ozone overburden is mostly repeatable from year to year with variability becoming a source of noise over a 15+ year record. There is of course an error related to long-term secular change in ozone. The drift error at 330 nm is related only to the mean change in total column ozone over the OMI record multiplied by the ozone sensitivity ratio between the shorter ozone-absorbed wavelength (used in retrievals) and 330 nm, approximately 10x. And 330 nm is merely the shortest wavelength we consider. As indicated in Section 5.5, our evaluation of the wavelength-dependent degradation is based on 340 nm and longer.

We do not claim the OMI radiometric changes are spectrally flat. This is rather unlikely. Instead, we are saying that the wavelength-dependent response change determined through the ice radiance technique is statistically consistent with zero (see Fig. 10). We will add a quantitative assessment of Fig. 10 to demonstrate this. As the reviewer correctly observes, the ALU1 change shown in Fig. 4 provides strong circumstantial evidence for spectrally flat sensor degradation. The small change seen in that figure is primarily a result of folding mirror degradation, an element not present in the Earth radiance measurements. An unfortunate consequence of the OMI design is the lack of a means to directly measure sensor change that includes the primary telescope mirror. What we can do is argue that this mirror's optical degradation as a result of photopolymerization is likely much less than that of the folding mirror because the latter sees a significant number of photons shortward of 300 nm and the former does not. If we use the ALU1 change as an upper bound to optical degradation at ozone-absorbed wavelengths, the resulting change in the measurement vector used in total column ozone retrievals is less than 0.1% over the mission (1% change between 250 nm and 500 nm; 317/331 nm pair used for ozone retrievals). We will include these additional points in the Section 5.5 discussion.

**Response to specific comments**

| Item | Referee comment | Author's response |
|---|---|---|
| Section 3.5 | "In addition, a static irradiance measurement used over a 17 year mission ignores the subtle changes in the solar output, an effect that could enter the L2 products in the long term." Can we really assume that the solar variability over a timescale of 17 years is negligible in terms of signal variation observed (in particular in the UV)? | We will add a statement on the stability of the Sun relative to our analysis method. The analysis was based (smoothed) over the entire (continuum) wavelength range and not based on variable Fraunhofer lines. |
| Section 4.5 | On flagging: Can you confirm then that a pixel qualification using originally 31 categories have been mapped down to 3 – and finally to only 1 in the end – with RTS being separated out? Was this mapping unique or were there some ambiguities to overcome? | The original 32 possible categories was an approach which did not work very well and was not user friendly. We indeed reduced the criteria that then result in one flag. |
| Section 5.3 | on relative irradiance: I would consider it clearer for the reader to talk about the diffuser BSDF – after having properly defined it – and its correction (which changes over time as a function of azimuth angle, elevation and time). So I would consider to replace "relative irradiance" with "diffuser BSDF" variation/correction. | On the fully integrated instrument, the diffuser BSDF cannot be measured separately from the rest of the optics and electronics. This is why we chose this terminology. We will consider alternate names for this sensitivity |
| Section 5.4 | The choice of the normalisation point is an extremely sensitive and delicate matter for deriving such a degradation correction. First of all because data at the day of the launch (as "start of the mission") cannot be used. But second also since the selected normalisation point (and its inherent biases) can significantly amplify biases in the normalised time series of correction coefficients for later periods.
• So what is characterized here as start of the mission? Ideally this should be the first irradiance measurement of the instrument, which can be solidly and fully calibrated (irrespective of commissioning periods or SIOV).
• On the other hand, various normalisation spectra should be tested to find out the sensitivity of the choice of the reference spectrum on the degradation correction coefficients. Has such a sensitivity test been carried out?
• Again I find the assumption on a completely stable sun over a 17-year period a bit tricky without further qualifications, in particular in the UV. | We will add which orbit we have used as start of the mission. It is an orbit at nominal operational temperature after the thermal testing at the beginning of the mission.
We will address the question of the Sun's stability relative to our analysis method. We have not performed a sensitivity analysis as such. However, per year references were used and together with the binning in wavelength dimension the random fault of a specific measurement should not translate into a bias which is significant compared to the uncertainty associated with the absolute radiometric calibration from on-ground calibration. |

| Item | Referee comment | Author's response |
|---|---|---|
| Section 5.5 | There seems to be a systematic dependency (at first order) of the degradation over the rows with higher degradation for the middle rows and lower for the edges. Is this potentially a systematic effect? | Yes, this is what we observe. The higher degradation might be related to the row anomaly and caused by higher exposure. |
| Section 6.2.2 | On the wavelength temperature correction: I would assume that the dependency of the dispersion on OPB has been measured on-ground. How do the results obtained in this study compare to the on-ground measurement temperature dependency of the spectral calibration stability? | As far as we are aware no usable temperature dependence measurements from on-ground are available. Collection 3 already used in-flight measurements for this. |
| Section 6.5 | On the "transient" signal flagging. How often are pixels flagged for this "transient events"? Can some statistics be provided, and are these events unknown in their nature/origin, and therefore not categorized as any of the pixel effects before? Only in the next section it becomes clear that cosmic particle impact is one of those transient effect. So a list of potential causes would set the scene here. | Agreed, we can change the order of the text here and provide some typical numbers. |
| Section 6.6 | High latitudes are of course also very significant regions of cosmic particle impact. Here only the (important) SAA area and its evolution is shown and discussed. I would assume that transient effects also accumulate and are accounted for at a global scale (ie including high latitudes). Can you confirm? | This is correct, the transient flagging is performed for all measurements. The SAA is the most likely region for transients and is therefore flagged separately (even if no transient is detected for a specific pixel). |

**Response to editorial comments**

| Item | Referee comment | Author's response |
|---|---|---|
| All | Generally, I think it would be very helpful to point to specific section in the supplement (ATBD) document, which is referred to throughout the paper at a time. This will help the reader (in particular the not so expert ones) to find their way through the vast amount of supplemental information provided in the ATBD (naturally not all relevant to the scope of this paper). | We will point to the relevant sections in the ATBD. |
| All | Generally, on figure captions: Captions often refer to top/down panels where there are only left/right panels | This will be corrected. |
| Section 2.1 | OB, or OBP or OPB? | It should be OPB. We will check that this abbreviation is used consistently. |

| Item | Referee comment | Author's response |
|---|---|---|
| Section 2.1, l. 132 | "it was observed that the duty cycle of the PWM of the UV detector heater occasionally dropped to zero,..". Can a concrete date be added to this? | We will add the date (beginning of October 2017). |
| Section 3.3 | It might be worth mentioning in this context that product format porting and restructuring to netcdf is part of a wider effort to streamline the product format of instrument of that type (GOME, SCIA, GOME-2, S5p, S5 and the future S7) and with the same AC and CLIM community in terms of output format (netcdf CF-standard) | We will add a comment on the effort to streamline product formats. |
| Section: 4.1 | So what is the correct value then? Since only the "erroneous" conversion is reported. | We will add an explanation. |
| Section 4.2, l. 282 | Check sentence | We will rewrite this and the following sentence. |
| Figure 19 | The terminology difference between "terrain height" and "surface altitude" is nowhere explained. | Will add an explanation. In both cases it's the height with respect to the reference geoide. |

**Detailed changes**

**List of changes to version**

The line numbers are with respect to the version reviewed by the referees.

| Item | Change | Response to comment of Referee # |
|---|---|---|
| Line 6 | Replaced "for many years more." by "until the expected decommissioning of its platform Aura in 2025." | #1 |
| Line 9 | Added: "and reprocess the data of the entire mission up to now." | #1 |
| Line 18 | Added: "TES was decommissioned on 31 January 2018 and HIRDLS ceased working on 17 March 2008." | #1 |
| Line 33 | Added: "during about 15 orbits." | #1 |
| Line 44 | Added: "For the calculation of the Earth's reflectance it is important to note that not all optical elements employed for radiance measurements are also included in the optical path for irradiance measurements: The first telescope mirror is bypassed by a folding mirror which directs the light from the solar diffuser to the second telescope mirror." | #1 |
| Line 49 | Added:" The former convention is slightly misleading as the VIS band also detects light in the ultraviolet." | #1 |
| Line 67 | Replaced "trend and calibration monitoring system" by "trend monitoring and calibration system" | #1 |
| Line 132 | Added: "(October 2017)" | #3 |
| Line 151 | Replaced "been" by "be" | |
| Line 171 | Added: "For the background correction the previous 24 hours of background data are aggregated." | #2 |
| Line 176 | Added: "using also the instrument configuration identifiers (ICID)." | #2 |
| Line 206 | Added: "The new format and structure of the OMI products is in line with the effort to streamline the product formats of similar instruments such as GOME, SCIAMACHY and the future missions Sentinel 5 and Sentinel 7." | #3 |
| Line 237 | Replaced "CKD" by "" calibration key data (CKD)" | #1 |
| Line 244 | Replace "alternative irradiance product is generated that consists of the…" by "alternative irradiance product is generated in a separate post-processor. It consists of the.." | #2 |
| Line 253 | Added: "The solar output varies about 0.1% between minimum and maximum of a solar cycle, see  Marchenko et al. (2016). " | #3 |
| Line 266 | Added: "for both UV and VIS." | #3 |
| Line 269ff | Replaced "For the collection 4 L01b processor, the ADC conversion factor is corrected to the conversion factor that was established during ground testing of the detection sub-system. The voltage-to-charge conversion factor is adjusted with the inverse of the change to the ADC conversion factor, so that the overall calibration is not changed. " by "For the collection 4 L01b processor, the ADC conversion factor was changed to the conversion factors that were established during ground testing of the detection sub-system. This calibrated conversion factor is 2910.4 DN/for UV and 2905.6 DN/for VIS. Accordingly, the | #3 |

| Item | Change | Response to comment of Referee # |
|---|---|---|
| | voltage-to-charge conversion factors were adjusted with the inverse of the change to the ADC conversion factor, so that the overall calibration is not changed." | |
| Line 282 | Replaced "In collection 3 a static value for each of the 4 gain ratios were used that were calibrated using year 2005 in-flight measurements. The collection 4 a temporal axis was included because the gain ratios are now known to drift in time. " by  "In collection 3 a static value was used for each of the 4 gain ratios. These values were derived from in-flight measurements from the year 2005. The gain ratios drift over time and in collection 4 the drift is corrected by adding a temporal dimension to the calibration key data." | #3 |
| Line 344 | Added: "An increased.." | #2 |
| Line 346 | Rephrased to  "should be the same in both the radiance measurement and its associated background. | #2 |
| Line 349 ff | Rephrased to "Therefore, a comparison between the expected noise and the observed noise of a pixel is sufficient to determine if a pixel suffers from RTS on this timescale. The expected noise is part of the L01b product and consist of the sum of read-out noise and shot noise. The observed noise is the temporal variance calculated from the ca. 800 dark frames accumulated in a day." | #2 |
| Line 370 | Added: "The wavelength annotation and corrections for wavelength shifts are described in Sect. 6.2." | #2 |
| Line 394 | Replaced "QVD diffuser" by "quasi volume diffuser (QVD)" | #1 |
| Line 396 | Replaced "The ALU1 regular diffuser is used once a week, and the ALU2 backup diffuser once a month; " by "The two aluminum surface reflection diffusers (ALU1 regular and ALU2 backup) are used once a week and once a month respectively;" | #1 |
| Line 403 | Replaced "and differs per their exposure ratio" by "differs according to their exposure ratio." | #2 |
| Line 415 | Added: "the Sun and the Earth" | #2 |

[revised manuscript text omitted]

**Changes to Figures and captions**

The numbers refer to the old (new) figure numbers

| Item | Change | Response to comment of Referee # |
|------|--------|--------------------------------|
| All | Refer to panels (a) and (b) where top/bottom or left/right panels were mentioned | All |
| Fig. *New* (9) | Added new figure and caption. | #3 |
| Fig. 14 (15) | Added: "The maximum shift of 140 pm corresponds to a shift of 0.42 detector pixels in UV1." | #2 |
| Fig. 15 (16) | Added: "A shift of 30 pm corresponds to 0.21 detector pixels in UV2." | #2 |
| Fig. 19 (20) | Replaced "the surface altitude for the same orbit in" by "the same for" | #3 |
| Fig.21 & 24 (22& 25) | Changed the y axis of the lower plots to be the same for all three bands. Changed to plot to make it more readable. Rephrased caption to refer to plots within the panels. | #1 |

Added specific sections when referring to the OMI collection 4 ATBD.

**Spelling changes/ typos/house style**

- Use American spelling consistently.
- Replaced OBP by OPB.
- Use solar instead of Solar.
- Changed date format to DD month YYYY
- Consistent use of level 0/1,  L0/L1b and L01b
- Consistent use of data in plural form ("data are..")

**Other changes**

Added other affiliations and corrected name for P. Levelt.